# Pressure encryption toward physically uncopiable anti-counterfeiting

Dianlong Zhao[1,2,8], Shunxin Li[1,8], Yang Su[3,8], Jiajun Qin ®[4,8], Guanjun Xiao ®[1] ✉, Yuchen Shang[1,5], Xiu Yin[1], Pengfei Lv[1], Feng Wang[1], Jiayi Yang[1], Zhaodong Liu[1,5], Fujun Lan ®[6,7], Qiaoshi Zeng ®[6,7], Lijun Zhang ®[3] ✉, Feng Gao ®[4] ✉ & Bo Zou ®[1] ✉

Current optical anti-counterfeiting technologies are mainly limited to materials with multicolor emissions, where the encryption method is only through photoexcitation. It brings about a huge risk for counterfeiting once these materials are reproduced. Here, we introduce a robust pressure encryption as the pressure engineering secret key to strengthen current optical anti-counterfeiting technique from pressure-induced emission luminogens. Through loading different pressures, the initially non-emissive 0D hybrid halide ($C_7H_{11}N_2$, 4DMAP)$_2$ZnBr$_4$ shows at least 8 different distinct bright emission colors. These color changes are attributed to controllable tuning of charge transfer and local excitation implemented by pressure treatment. Moreover, the unique color tuning through pressure loading, randomized distribution of the fluorescent particles, as well as designated micro-nano patterns greatly enhance the security capability of current visual information encryption, which serves as the triple-level physically uncopiable optical anti-counterfeiting technique. Our work provides a promising strategy of materials-by-design for high-performance anti-counterfeiting, imaging and information storage applications.

Optical encryption enables anti-counterfeiting technology with high security, which is highly promising in various fields such as banknotes, authenticity guarantee of luxury goods, and information security[1–3]. Specifically, most optical encryption technique relies on the uniqueness of the luminescent materials, such as the intensity, color, and lifetime features of the photoluminescence (PL) response. The organic metal halide hybrid materials (OMHHs) are the successful examples due to the easy cheap processing, abundant color tuning and highly luminescent properties[4–7]. However, these materials suffer from huge risk of being counterfeited once their compositions are revealed. It is urgent to add new optical encryption strategies in the current optical encryption to maintain or even strengthen the high anti-counterfeiting security, which contributes to addressing the increasing global prevalence of counterfeiting[8–10].

Pressure engineering appears to be a promising optical encryption strategy owing to the new dimension of materials tuning such as bandgap tuning, piezochromism and pressure-induced emission (PIE) harvesting, etc[11–14]. Especially for the PIE harvesting in metastable

[1]State Key Laboratory of High Pressure and Superhard Materials, College of Physics, Jilin University, Changchun 130012, China. [2]School of Physics and Materials Engineering, Dalian Minzu University, Dalian 116600, China. [3]School College of Materials Science and Engineering, Jilin University, Changchun 130012, China. [4]Department of Physics, Chemistry and Biology (IFM), Linköping University, Linköping 58183, Sweden. [5]Synergetic Extreme Condition User Facility Jilin University, Changchun 130012, China. [6]Center for High Pressure Science and Technology Advanced Research, Shanghai 201203, PR China. [7]Shanghai Key Laboratory of Material Frontiers Research in Extreme Environments (MFree), Institute for Shanghai Advanced Research in Physical Sciences (SHARPS), Shanghai 201203, PR China. [8]These authors contributed equally: Dianlong Zhao, Shunxin Li, Yang Su, Jiajun Qin. ✉e-mail: xguanjun@jlu.edu.cn; lijun_zhang@jlu.edu.cn; feng.gao@liu.se; zoubo@jlu.edu.cn

states, it has been realized in halide perovskites or derivative hybrid halides, which inspires the field of advanced optical encryption via pressure engineering[15–21]. For instance, in (PEA)$_2$PbCl$_4$, pressure engineering enabled the retention of warm white-light emission, with wavelength tuning ranges extending up to 80 nm[20]. Similarly, in (NAPH)$_2$PbCl$_4$, the quenched warm white-light emission was achieved, exhibiting a tuning range of 70 nm[16]. However, both materials only showcased two-color emission tuning after applying different pressures. (PEA)$_2$PbCl$_4$ transitioned from warm white-light to cold white-light emission, while (NAPH)$_2$PbCl$_4$ shifted from cold white-light to warm white-light emission. Furthermore, the objects of PIE luminogens for above-mentioned reports exhibited initially weak emission. In the scenario of future advanced optical encryption with pressure, multicolor PIE luminogens (PIEgens) are expected in one material via loading different pressures, which serves as the pressure engineering secret key (PESK). However, its development is hindered by the lack of highly distinct color changes for most materials after loading high pressures.

In the present work, we successfully realize PESK from PIEgens in an originally non-emissive OMHH (4DMAP)$_2$ZnBr$_4$, by taking advantage of its unique charge-transfer exciton emission. After loading different pressures on this material, a series of different bright PL colors are revealed from dark state "0" to bright state "1", ranging from deep blue to orange-red. As a result, at least 8 different distinct colors with varying emission intensities and lifetime are achieved, which suggests the high security level of our PESK. In situ synchrotron XRD, infrared (IR), absorption, HRTEM and pair distribution function (PDF) experiments elucidate that multicolor emissions are attributed to the Urbach tail induced by amorphous degrees implemented by pressure engineering. The time-dependent density functional theory (TD-DFT) calculations reveal that the PIE is highly associated with the mixed localized excitation of 4DMAP$^+$ and halogen-to-ligand charge transfer states from the [ZnBr$_4$]$^{2-}$ to the 4DMAP$^+$. The steric hindrance effect together with enhanced hydrogen bond interaction from organic aromatic 4DMAP$^+$ is responsible for the retention of bright emissions. Combined with micro-nano patterning structures, diverse patterns of QR codes, school logos, cats, ginkgo leaves, pandas and butterflies etc. are successfully achieved. Particularly, the randomized distribution of fluorescent particles within the designated cat patterns is established, resulting in triple-level and physically uncopiable optical anti-counterfeiting from PIEgens. It considerably enhances the ability of naked-eye recognition and visual information encryption. The study offers a robust strategy for the rational structural design of novel metastable states with high-performance optical anti-counterfeiting and information storage applications.

## Results and Discussion

In order to realize PESK, we introduce the low-dimensional OMHHs with multicolor emissions owing to their integrating properties derived from organic and inorganic components[5]. We design and synthesize a zero-dimensional (0D) compound (4DMAP)$_2$ZnBr$_4$ with unique space group *C*2/*c* pseudo-layered structure (Supplementary Fig. 1a). Band projection in Supplementary Fig. 1b shows a component-dependent electronic structure of (4DMAP)$_2$ZnBr$_4$ under ambient conditions. It indicates that the conduction band minimum (CBM) is mainly dominated by 4DMAP$^+$, and the valence band maximum (VBM) is contributed by the mixed [ZnBr$_4$]$^{2-}$ and 4DMAP$^+$. Frontier orbital analyses further suggest the emission features of intramolecular charge transfer (ICT) states in 4DMAP$^+$ and halogen-to-ligand charge transfer (HLCT) states between [ZnBr$_4$]$^{2-}$ and 4DMAP$^+$ (Supplementary Fig. 1c, d)[22]. This unique band structure provides a possible color tuning strategy by adjusting the distance between [ZnBr$_4$]$^{2-}$ and 4DMAP$^+$ though loading pressure. Therefore, the PESK is theoretically achievable.

Now we check our analysis by characterizing the PL spectra under different pressures. Under ambient conditions with one standard atmosphere (1 atm), the PL emission signal of the compound (4DMAP)$_2$ZnBr$_4$ is barely seen (Supplementary Fig. 2a). Upon increasing the pressure, the PL spectra becomes visible with the intensity increasing accordingly. There are always two peaks when the pressure is high and the peak position shifts with the pressure. Interestingly, after removing the high pressure, the compound becomes emissive under ambient conditions. As shown in Supplementary Fig. 2f, when the loaded high pressure is up to 21.5 GPa, the compound exhibits bright green color under ambient conditions, suggesting that the originally non-emissive (dark state "0") compound changes to highly emissive (bright state "1").

A series of different high pressures ranging from 7.1 GPa to 35.6 GPa are tested, resulting in bright PL multicolor emissions after releasing the pressure with peak shifting accordingly (Fig. 1a). Figure 1b further shows the monolithically decreased PL emission energy with the increased loaded pressure, suggesting the excellent encryption strategy through loading high pressure. Moreover, the color gamut tuning range is wide, ranging from deep blue to orange-red light, exhibiting easily distinguished distinct colors even for the naked eyes (Supplementary Fig. 3). We should point out that it is the first time we reveal bright multi-color tunability through loading pressure and releasing it, rather than the conventional ways by changing material component, tuning excitation wavelength or introducing heating. Furthermore, variations in PL intensity and lifetime after loading different pressures contributed to the increased multi-dimensionality of the PESK (Fig. 1c and Supplementary Fig. 4). Thus, we provide a new strategy with high encryption level for the PESK application from PIEgens.

For the real PESK application, the stability of the color emission after loading the pressure is an important factor. According to the HLCT nature between [ZnBr$_4$]$^{2-}$ and 4DMAP$^+$, the color change with loaded pressure is associated with the distance herein. Thus, we evaluate the color stability by considering the hydrogen bond binding energy and the steric hindrance effect after loading pressure. The calculated hydrogen bond binding energy increases 1.52 eV after the pressure was completely released as compared with the pristine material without loading pressure. In addition, the calculated steric effect index (SEI) of 4DMAP$^+$ was 2.45 (Supplementary Fig. 5). The large hydrogen bond binding energy increase together with high SEI value implies the excellent retention of emissions after loading pressure. The experimental stability tests further verify our analysis. Specifically, the PL peak can remain stable with the PL intensity degrading slightly for even up to one month after pressure is completely released to ambient conditions (Fig. 1d and Supplementary Fig. 6). Additionally, this degradation of PL intensity can be recovered by loading the pressure again (see the cycling stability test in Fig. 1e and Supplementary Fig. 7). Indeed, the choice of the pressure transmitting might influence the degree of hydrostaticity in the compression, which, in turn, might determine a different outcome in the material response[23,24]. Usually at pressures up to 10 GPa, the pressure transmitting medium yields nearly hydrostatic conditions. We conducted the relevant experiments using silicone oil and ethanol/methanol with a volume ratio of 4:1 as the pressure transmitting medium to systematically examine the influence of the medium on the degree of hydrostaticity. The experiments were performed on samples R-5.2, R-4.1, R-3.1, R-2.1, R-1.1, and 0 (Supplementary Fig. 8a, b). Upon comparing the PL spectra from the two scenarios, it is evident that PIE is present in both cases, with no discernible difference in the spectra between them. Furthermore, we conducted a comparison of photoluminescence after loading high pressures up to 21.5 GPa. The results revealed minimal differences (Supplementary Fig. 8c). Unlike ABO$_3$-type perovskite structures, (4DMAP)$_2$ZnBr$_4$, as low-dimensional hybrid halides, do not strictly adhere to the Goldschmidt tolerance factor rules[25]. The incorporation

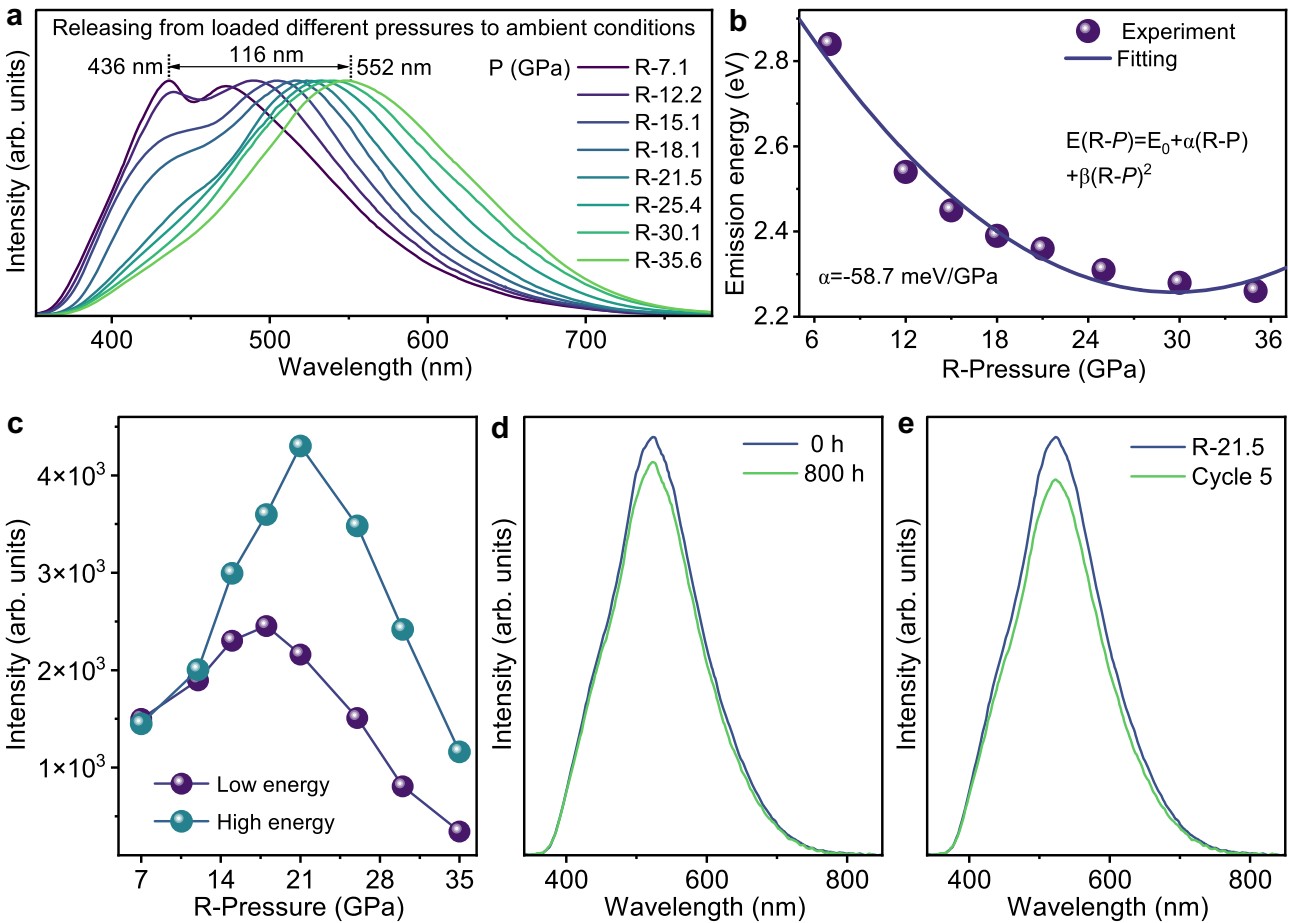

**Fig. 1 | Potential of (4DMAP)₂ZnBr₄ as the PESK application. a** Normalized PL spectra of (4DMAP)₂ZnBr₄ measured at 1 atm after loading different pressures. **b** Relationship between emission energy and released pressure from different pressure engineering points. **c** Intensity values for the PL signal of all the samples R-7.1, ..., R-35.6. **d, e** PL intensity of quenched (4DMAP)₂ZnBr₄ as a function of aging time and cycle numbers by releasing from pressure of 21.5 GPa, respectively.

of large condensed ring amine cations 4DMAP⁺ at the A sites is permitted. The limited space facilitates the formation of robust hydrogen bonds both among the cations and between these cations and the Br⁻ ions. Therefore, the (4DMAP)₂ZnBr₄ material exhibits a highly dense structure. As a result, the molecules of the pressure transmitting medium are unable to penetrate the material, merely providing a hydrostatic pressure environment on its surface, leaving these interactions unaffected. Therefore, the selection of the pressure transmitting medium does not significantly affect our experimental findings. Note that there exists a pressure threshold above 2 GPa, which the intensity gain of PIE is interrupted (Supplementary Fig. 8a, b). To investigate different kinetics during the pressure cycle and determine a different response in the sample, we conducted experiments with different compression rates (Supplementary Fig. 9). It is found that the compression rate below 1 GPa/1 min has a negligible effect on the emissions. However, when the compression rate exceeds 1 GPa/0.5 min, there will be a decrease in emission intensity, attributed to the enhanced amorphous degree[26]. Therefore, it is possible that different kinetics in the pressure cycle determines a different response in the sample.

To understand the PIE of the 0D halide compound (4DMAP)₂ZnBr₄, we conduct in situ high-pressure synchrotron XRD, absorption, infrared (IR) and TD-DFT calculations. With increasing pressure, all Bragg diffraction peaks shift to higher angles (Fig. 2a), exhibiting a reduction in both lattice distance and unit cell volume (Fig. 2b). Here, high pressure compresses the lattice structure, making it more rigid and effectively inhibiting molecular rotation. Thus, non-

radiative transitions are suppressed, resulting in enhancement of PL efficiency. The absorption edge exhibits a continuous redshift with increasing pressure, accompanied by the emergence of a Urbach tail localized states in the band tails (Supplementary Fig. 10a), which is attributed to the amorphous disorder in the aggregated compounds[27–29]. Remarkably, this amorphous aggregation maintains even after the pressure is released, as evidenced by the residual Urbach tail in the absorption spectra (Supplementary Fig. 10c–h). It rationalizes the unique PESK characteristics of our 0D compounds that the pristine non-emissive (4DMAP)₂ZnBr₄ becomes emissive at high pressure and remains even when the pressure is released. Under high pressure, the N–H stretching vibration ν (N–H) shows a redshift (10 cm⁻¹) from 3250 to 3240 cm⁻¹ at 6.8 GPa (Fig. 3a)[30]. Further results of redshift in the IR vibration verify the maintaining of the aggregation after high pressure is released (Fig. 3b and Supplementary Fig. 11a–f). Both the angle-dispersive X-ray diffraction (ADXRD) patterns and IR spectra confirm the amorphous nature of the aggregates, as evidenced by the weakened peaks and the broadened FWHM in Figs. 2c, 3c, respectively. This amorphous structure induces localized states in the band tails, known as the Urbach tail, which play a critical role in the observed emission behavior.

To elucidate the PESK properties of our 0D compound, the excited states simulations are calculated using TD-DFT calculations, as shown in Fig. 4a, b. Under ambient conditions, the distribution of electron-hole wave functions reveals emission features of intramolecular charge transfer and halogen-to-ligand charge transfer (HLCT). Under high pressure, the electron–hole overlap increased in the both

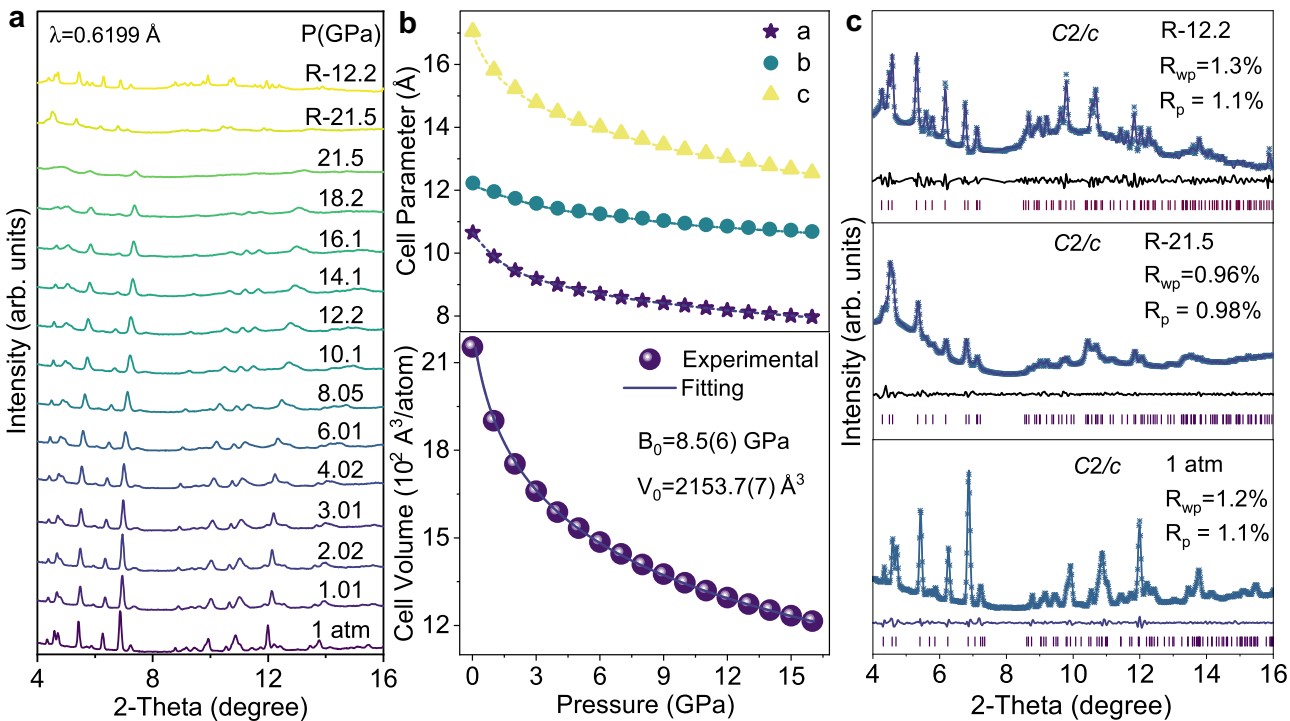

**Fig. 2 | Changes in ADXRD and lattice parameters during and after loading different pressures. a** Powder X-ray diffraction patterns representative of (4DMAP)$_2$ZnBr$_4$ under high pressure. **b** Experimental lattice constants (**a**–**c**) and volume of (4DMAP)$_2$ZnBr$_4$ as a function of pressures. The solid lines represent the Birch-Murnaghan EOS functions fitted to the measured P-V data. **c** Comparison of refinement results for (4DMAP)$_2$ZnBr$_4$ at 1 atm, R-21.5 and R-12.2 GPa, respectively. The purple bars indicate the positions of the refined Bragg peaks, while the black line represents the difference between the experimental (blue crosswire) and simulated calculated (purple solid line) diffraction profiles.

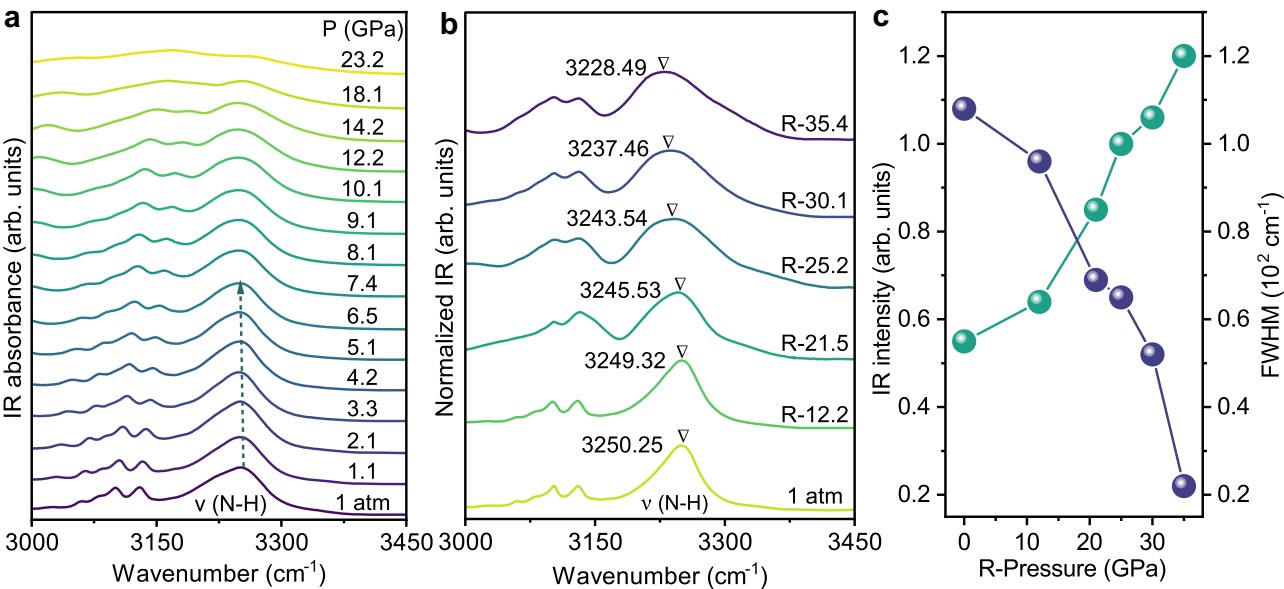

**Fig. 3 | Changes in IR vibrational spectrum during and after loading different pressures. a** Evolution of vibrational spectrum of (4DMAP)$_2$ZnBr$_4$ under high pressure. **b** Evolution of vibrational spectrum of (4DMAP)$_2$ZnBr$_4$ after decompression. **c** Intensity and full width at half maximum (FWHM) evolution of IR vibrational spectrum of (4DMAP)$_2$ZnBr$_4$ after decompression.

excited state and ground state of intramolecular 4DMAP$^+$ (Fig. 4b), thereby promoting radiative recombination of localized excitons. Moreover, the augmented distribution change is observed in the electron-hole wave functions between 4DMAP$^+$ and [ZnBr$_4$]$^{2-}$, owing to increased coupling between them, which facilitates the HLCT process. The charge transfer amount contributed by atoms has increased from 2.6% to 6.5% (Fig. 4a, b). Meanwhile, as depicted in Fig. 4c, the oscillator strength of the emission in the (4DMAP)$_2$ZnBr$_4$ increased significantly by a factor of 26 under high pressure as compared to that at 1 atm, which is in good agreement with observed emission enhancement. Therefore, under ambient conditions, the non-emissive behaviour of (4DMAP)$_2$ZnBr$_4$ could be attributed to the presence of isolated

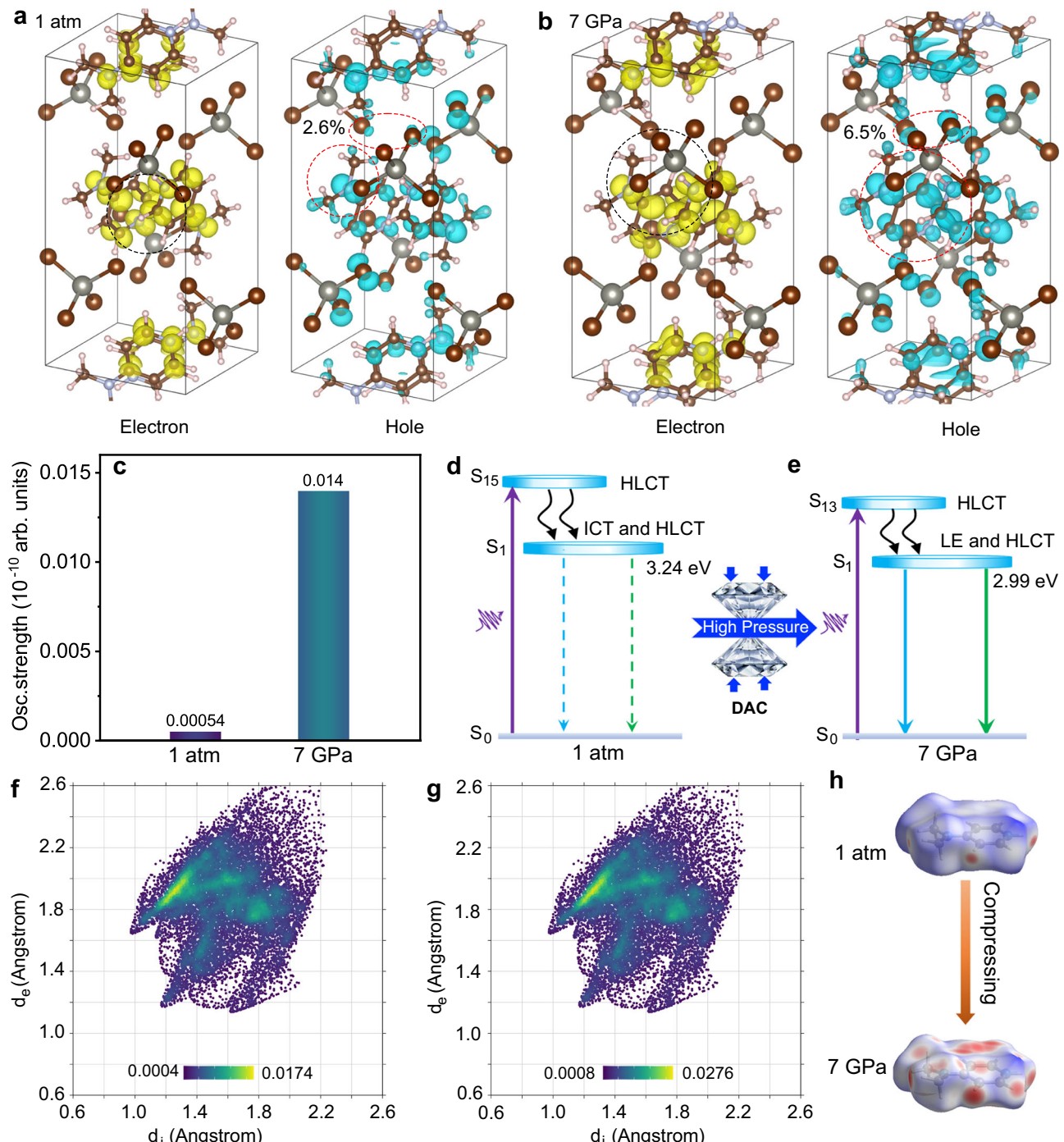

**Fig. 4 | Nature of PIE under high pressure. a, b** Calculated hole and electron wave functions of (4DMAP)$_2$ZnBr$_4$ at 1 atm and 7 GPa. Circles represent the modifications in the wave functions of the hole and electron, respectively, shown with isosurfaces (0.0012 au). **c** Calculated emission oscillator strengths of (4DMAP)$_2$ZnBr$_4$ at 1 atm and 7 GPa. **d, e** Schematic energy diagram of emissions under high pressure. **f, g** 2D fingerprint plots of calculated (4DMAP)$_2$ZnBr$_4$ structures at 1 atm and 7 GPa. **h** Hirshfeld surface analysis upon compression from 1 atm to 7 GPa.

rotating molecules and a high charge excitation energy. At high pressure, the bright dual emission in (4DMAP)$_2$ZnBr$_4$ could be attributed to the enhancement of mixed LE and HLCT emissions, which arise from increased hydrogen bonding interactions and a decrease in charge-transfer excitation energy from 3.24 to 2.99 eV (Fig. 4d–h and Supplementary Fig. 12)[31].

The increment of luminescence efficiency by loading high pressure is also revealed by Hirshfeld calculations (Fig. 4f–h)[32]. The aggregation of these 4DMAP$^+$ pyridines by intermolecular N-H⋯π hydrogen bonds

to form complex motifs could result in much slower recovery rates compared to the inorganic components, possibly accompanied by structural reconstruction during decompression[33]. In addition, calculated rotational single-point energies represent that the energy required for 4DMAP$^+$ rotation without hindrance is increased significantly under high pressure, and this effect persists even after the pressure is released (Supplementary Figs. 13, 14, Tables SI, SII). Note that obvious redshift in the IR vibration and amorphous ADXRD pattern upon decompression are retained under ambient conditions, which

coincides with the residual Urbach tail in the absorption spectra (Supplementary Fig. 10c–h). Therefore, the steric hindrance effect together with enhanced hydrogen bond interaction from organic aromatic 4DMAP$^+$ group would increase the potential energy barrier for the amorphous state transition, preserving PIE to atmospheric pressure.

To investigate the origin of multicolor emissions of dense (4DMAP)$_2$ZnBr$_4$, we conduct the absorption experiments, infrared (IR), pair distribution function (PDF), Structure factor S(Q), TD-DFT calculations, high-resolution transmission electron microscopy (HRTEM) and selected area electron diffraction (SAED). As shown in Fig. 5a, upon decompression from a series of high pressures to ambient condition, with higher loaded pressure, Urbach tail of the absorption is stronger, resulting in a decrease in the band gap (Fig. 5b). It indicates a more prominent aggregated state, which is in good agreement with more obvious redshift in the IR vibration (Fig. 3b). HRTEM, SAED and Structure factor S(Q) analyses in Supplementary Fig. 15 indicate that the compound undergoes a transition from a crystalline to an amorphous state. As shown in Fig. 5d, g, the pair distribution function (PDF) elucidates the presence of long-range ordered crystals and short-range ordered amorphous states, which is in accordance with calculated radial distribution function (RDF). The amorphous states exhibit shorter bond lengths, indicative of their densely aggregated states[34,35]. The properties of materials are determined by their structure, which is characterized through molecular dynamics simulations (Supplementary Fig. 16). As demonstrated in Supplementary Fig. 17, under high pressure, the application of pressure drives a transition from a dynamically disordered phase at ambient conditions to a statically disordered phase at high pressure. A similar mechanism occurs in (4DMAP)$_2$ZnBr$_4$ compared to ABO$_3$-type perovskite structures[36,37]. However, it was observed that the rotation angle of cations 4DMAP$^+$ is limited. Simultaneously, the distances between cations as well as between these cations and Br$^-$ anion gradually decrease. The rigid, large condensed ring amine cations 4DMAP$^+$, featured plastic deformation and distorted structure within the (4DMAP)$_2$ZnBr$_4$ system (Supplementary Fig. 17b, d–f). After high-pressure engineering treatment, the steric hindrance effect resulting from the irreversible plastic deformation, along with distorted large condensed ring amine cations 4DMAP$^+$ and the enhanced hydrogen bonding interaction are responsible for maintaining the amorphous phase under ambient pressure (Supplementary Fig. 17c, g–i). The calculated oscillator strength in Fig. 5c reveals that partially amorphous states exhibit the strongest emission efficiency because of its enhanced rigidity. Further increasing the degree of amorphous state can still maintain relative strong emission while the intensity declines a little bit because of non-radiative transitions. After the pressure is completely released, the increased hydrogen bonding interactions are evidenced by Hirshfeld surface analysis and 2D fingerprint plots (Fig. 5e, f, h, i). To elucidate the photophysical mechanism of emission in our compound, the excited states simulations of decompressed (4DMAP)$_2$ZnBr$_4$ are conducted by utilizing TD-DFT calculations. As depicted in Fig. 5j–l, the transition characteristics of LE and HLCT are monitored by the calculation of transition density matrix (TDM) between 1 atm and recovered from 30 GPa. The controllable tuning of HLCT and LE implemented by pressure treatment are responsible for the ultimate multicolor emissions.

Motivated by the PIE efficiency enhancement and the multicolor emissions of our compound, we further explore its potential applications in anti-counterfeiting and information storage. A multi-level physically uncopiable optical anti-counterfeiting modal paradigm is designed (Fig. 6). The PESK provides the capability to select unique fluorescence wavelengths from the multicolor emissions, thus demonstrating the first-level optical anti-counterfeiting mode (Fig. 6a). The 0D compounds enable the easy fabrication of micro-nano patterns, which constitutes the second-level optical anti-counterfeiting mode (Fig. 6b). For instance, we successfully demonstrate various micrometer-scale patterns, such as QR codes, school logos, "JLU" text, cats, ginkgo leaves, panda and butterflies (Supplementary Fig. 18)[38]. Here, the high uniqueness and randomness increased the difficulty of counterfeiting by utilizing fluorescent wavelengths. Additionally, the fluorescent patterns can carry encrypted information, such as microscopic QR code labels. These labels can be deciphered using microscopic fluorescence imaging, enabling information retrieval from the embedded website information. In the case of QR code labels displaying green and cyan-blue emission, users can access the homepage of Jilin University and Linköping University with a phone (Fig. 6c).

Although the anti-counterfeiting mode at the second level can be designed to be a covert and challenging system, yet it still presents a decoding risk for potential counterfeiters through the utilization of specific micro-nano structures. To further increase the anti-counterfeiting level, we develop a robust triple-level, uncopiable anti-counterfeiting mode by regulating the randomized distribution of fluorescent particles within the designated patterns (Fig. 6d). This indicates that each micro-nano pattern possesses a distinctive fluorescent feature, rendering it highly arduous to counterfeit or imitate. Uncopiable characteristics including different sizes, geometric dimensions, deposition location, and PL intensity of fluorescent particles, greatly prevent being accurately copied. As shown in designated cat patterns, although two cats exhibit the similar outline and consistent fluorescence peak position, their internal structures are completely distinct because of their uncopiable characteristics. Consequently, the counterfeit "cat" can be easily distinguished from genuine "cat" by the identification of the internally fine fingerprint information. Overall, the implementation of a triple-level and physically uncopiable anti-counterfeiting mode significantly enhances the anti-counterfeiting capability for naked-eye recognition and visual information encryption.

In summary, we successfully develop triple-level and physically uncopiable optical anti-counterfeiting modal materials through combining PESK with micro-nano patterning structures from PIEgens. The optical anti-counterfeiting materials featured multicolor emissions, including orange-red, yellow, green, cold white light, cyan-blue, and deep-blue emissions, etc., which are attributed to controllable tuning of HLCT and LE implemented by pressure treatment. The TD-DFT calculations elucidate that enhanced PIE is highly associated with the mixed localized excitation of 4DMAP$^+$ and halogen-to-ligand charge transfer from the [ZnBr$_4$]$^{2-}$ to the 4DMAP$^+$. The steric hindrance effect together with the enhanced hydrogen bond interaction from organic aromatic 4DMAP$^+$ group contributed to the retention of emissions through pressure processing. The calculated binding energy of hydrogen bond demonstrated that the released structure remains in a high-energy state that is difficult to recover. The triple-level and physically uncopiable optical anti-counterfeiting modal PIEgens greatly prevent them from being accurately copied because of regulating the randomized distribution of fluorescent particles within the designated patterns. Furthermore, multi-color combination of information encryption from PIEgens would highly enhance the anti-counterfeiting capability. This work provides a robust strategy for the rational structural design of novel metastable materials with high-performance applications in anti-counterfeiting, naked-eye recognition imaging, and physically uncopiable visual information encryption.

## Methods
### Materials source
All chemicals used to prepare the products were procured from commercial suppliers: 4-dimethylaminopyridine, (4DMAP, Sigma, Aldrich, 98%), zinc bromide (ZnBr$_2$, Strem Admas, 99.99%), N,N-dimethylformamide (DMF, Sigma-Aldrich, 99.9%), n-octylamine (Sigma-Aldrich, 99%), hydrobromic acid (HBr, Aladdin, 48% in water), n-hexane (Sigma-Aldrich, 99.9%), and methanol (Sigma Aldrich, >99.8%).

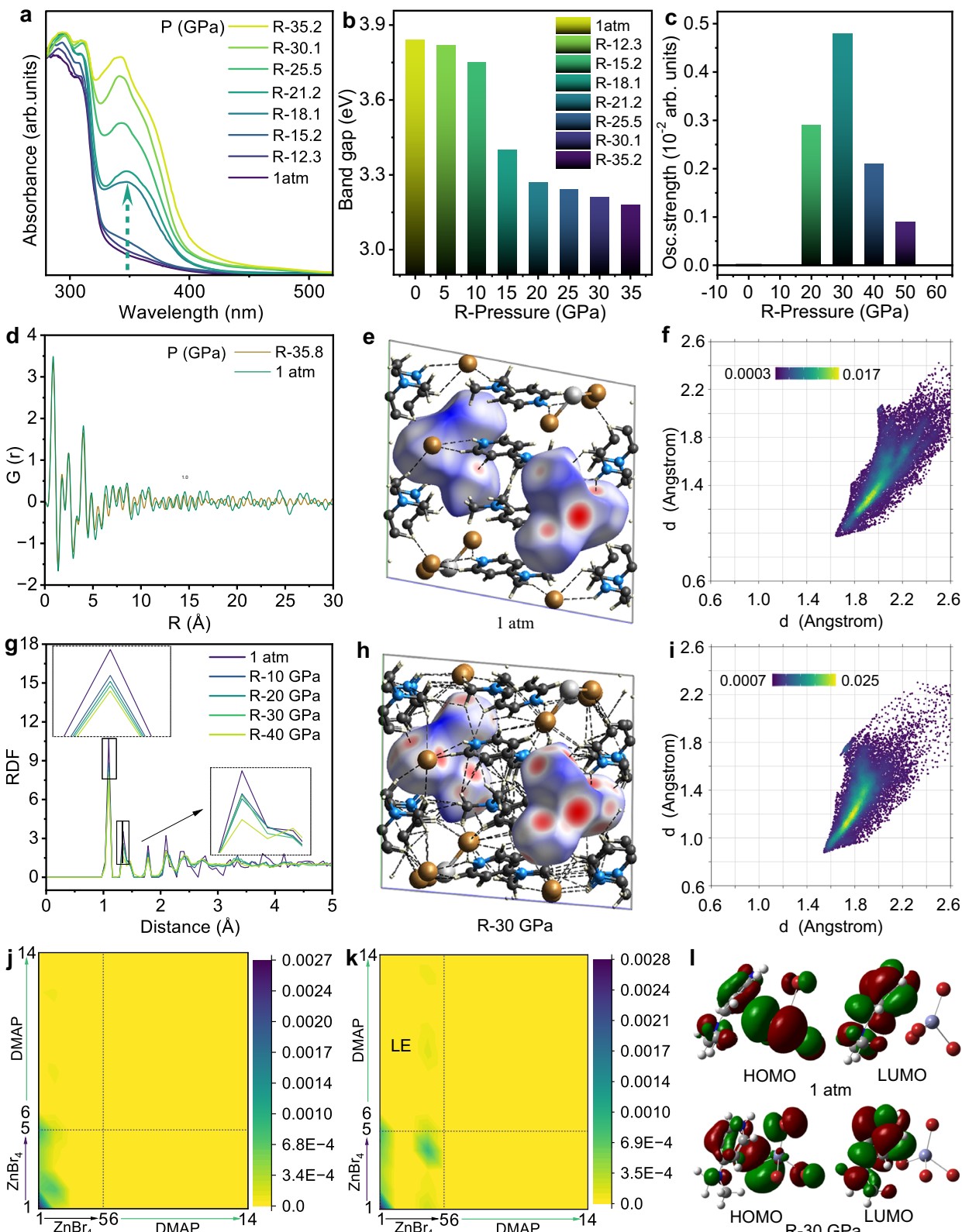

**Fig. 5 | Investigation of the origin of multicolour emissions of (4DMAP)$_2$ZnBr$_4$.**
**a** Comparison of absorption between 1 atm and recovered from different pressure engineering points. **b** Comparison of band gap between 1 atm and recovered from different pressures. **c** Calculated oscillator strengths between 1 atm and recovered from different pressure engineering points. **d** Comparison of PDF profiles, G(r) between 1 atm and recovered from 35.8 GPa. **e**, **f** Hirshfeld surface analysis and 2D fingerprint plots of calculated (4DMAP)$_2$ZnBr$_4$ structures at 1 atm. **g** Comparison of calculated RDF profiles, including 1 atm, R-50, R-40, R-30, R-20, respectively.
**h**, **i** Hirshfeld surface analysis and 2D fingerprint plots of calculated (4DMAP)$_2$ZnBr$_4$ structures at R-30 GPa. **j**, **k** Calculated transition density matrix (TDM) at 1 atm and R-30 GPa. Both horizontal axis $x_i$ and vertical axis $y_i$ run over all the labels of non-hydrogen atoms. **l** HOMO → LUMO in the electronic transitions at 1 atm and R-30 GPa.

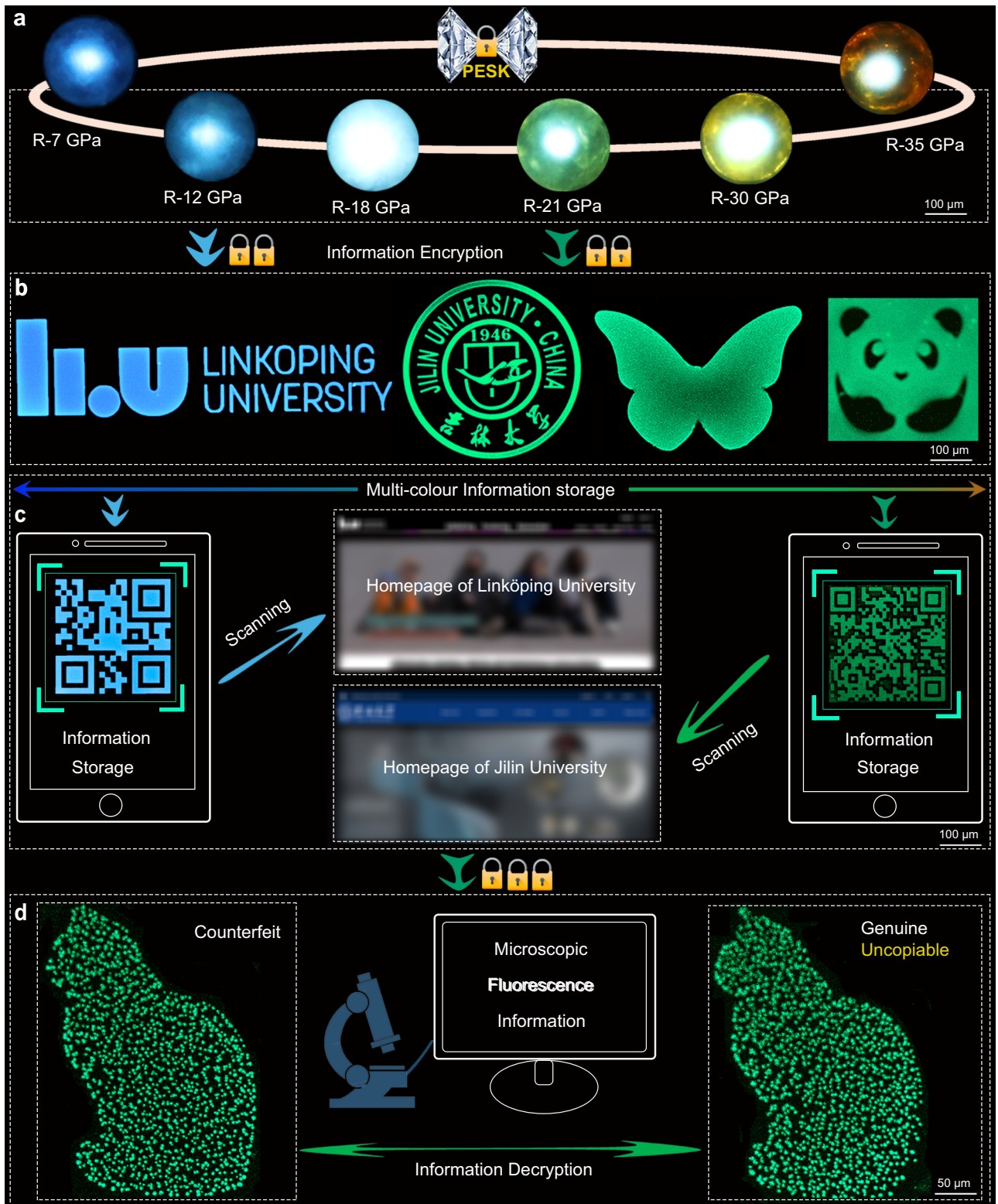

**Fig. 6 | Application of imaging, information storage and anti-counterfeiting based on PIEgens obtained upon decompression from Walker-type large-volume press. a** Quenched photographs from different pressures for the first-level optical anti-counterfeiting mode with PESK. **b** Second-level optical anti-counterfeiting mode based on micro-nano patterns including school logos, butterflies, panda, etc. **c** Information storage with accessing the homepage of Jilin University and Linköping University by scanning the fabricated QR code. **d** Triple-level uncopiable anti-counterfeiting mode and information decryption with distinguishing genuine cat from counterfeit.

## Sample preparation of $(4DMAP)_2ZnBr_4$

In the typical synthesis, 0.2 mmol 4DMAP, 0.1 mmol $ZnBr_2$, 1 ml HBr, and methanol (3 ml) were passed into a 30 ml conical flask. The mixture was stirred at room temperature, until the solution was clear. The solutions were allowed to slowly evaporate at approximately 40 °C in an oven, resulting in the formation of colorless crystal blocks of $(4DMAP)_2ZnBr_4$. In addition, pressure-quenched samples were obtained by Walker-type large-volume press.

## In situ high-pressure optical experiments

Firstly, the thickness of the T301 steel plate was pre-pressed to about 45 micrometers using a diamond anvil cell (DAC). Then a laser was used to drill a hole with a diameter of one-third of T301 steel plate, where the hole is used as the sample chamber. In the PL experiment, the powder samples are uniformly dispersed in silicone oil, which acts as a pressure-transmitting medium to establish a hydrostatic environment. Similarly, in the UV-vis absorption experiment, the suppressed transparent bulk thin layer sample is immersed in silicone oil, creating a hydrostatic pressure environment around the sample. At the same time, Ruby is used for pressure calibration inside the hole. It should be noted that during the infrared experiment, due to the peaks associated with silicone oil, KBr is used as the pressure transmitting medium. PL measurements were performed using a 355 nm UV DPSS laser, with a power output of 4.0 mW. The high-pressure evolution of the steady-state PL spectra was measured using a modified spectrophotometer (Ocean Optics, QE65000). The laser beam passed through a tunable filter and was focused onto the sample using a 20 μm spot from a 20×UV Plan apochromatic lens. The products' PL micrographs were captured using a camera (Canon EOS 5D Mark II) mounted on a microscope (Eclipse TI-U). The photographs were recorded by the camera under identical conditions, including exposure time and intensity. The absorption spectra were measured using a Deuterium Halogen light source in the region of exciton absorption band.

## Experiments of Walker-type large-volume press

High-pressure quench experiments were conducted at 12 GPa and 21 GPa by using a 10/5 (OEL/TEL = octahedral edge length of pressure medium/truncated edge length of anvil) cell assembly in a 10-MN Walker-type large-volume press at the State Key Laboratory of High Pressure and Superhard Materials, Jilin University[39]. The starting $(4DMAP)_2ZnBr_4$ powders were housed in a rhenium capsule and placed into $MgO$ sleeves within a $Cr_2O_3$-doped MgO octahedron. The pressure calibration was performed using the following approach: the resistance variations in ZnTe yielded the loads for 6.6 GPa, 8.9 GPa, and 12.9 GPa, whereas GaAs resistance provided the load for 18 GPa. A pressure-load curve for the 10/5 assembly was fitted, enabling the determination of the synthesized pressure[39]. Initially, the samples were compressed to predetermined pressures over a span of approximately 10 h at room temperature, followed by a gradual pressure release lasting roughly 10 h.

## Micro-Patterning Preparation

The $(4DMAP)_2ZnBr_4$ sample was loaded to 12 GPa and 21 GPa using a Walker-type large-volume press, followed by decompression to ambient conditions to achieve the desired luminescent properties. After pressure treatment, the modified sample was carefully removed from the press chamber and ground into a fine powder. The powder was then dispersed into an ethanol solution containing 1 wt% PVP to ensure uniform dispersion. The resulting dispersion was drop-coated onto the substrate and brought in contact with a mold, fabricated using laser direct writing or electron beam etching, to form a dispersion-substrate-mold system. This system was left in room temperature for 12 h to allow complete solvent evaporation. During evaporation, capillary forces drove the material to deposit onto the glass substrate, forming precise micron-scale patterns that replicated the mold design. Once the solvent had fully evaporated, the mold was carefully removed, leaving behind the patterned $(4DMAP)_2ZnBr_4$ on the substrate.

## In situ high-pressure synchrotron ADXRD measurements

The high-pressure in situ angle-dispersive XRD experiments were primarily conducted at the 4W2 HP-Station, located at the Beijing Synchrotron Radiation Facility (BSRF). $CeO_2$ was utilized as the standard sample for calibration purposes. The diffraction pattern was integrated into a one-dimensional profile using the Fit2D program. The structure factor S(Q) was performed at the Center for High Pressure Science and Technology Advanced Research, Pudong, Shanghai 201203, China. All the high-pressure experiments were conducted at room temperature. The third-order Birch-Murnaghan equation of state to fit the experimental pressure-volume (P-V) data of $(4DMAP)_2ZnBr_4$ as follows:

$$P(V) = \frac{3B_0}{2}\left[\left(\frac{V_0}{V}\right)^{7/3} - \left(\frac{V_0}{V}\right)^{5/3}\right]\left\{1 + \frac{3}{4}(B_0' - 4)\left[\left(\frac{V_0}{V}\right)^{2/3} - 1\right]\right\}$$

(1)

Given a parameter labeled as $V_0$ for zero-pressure volume, $B_0$ as the bulk modulus at ambient pressure, and $B_0'$ as the pressure derivative parameter. The isothermal bulk modulus $B_0$ was determined to be approximately 8.5 GPa.

## First-principles calculations details

The geometric optimizations and electronic excitation computations were performed using density functional approach as implemented in the CP2K codes[40]. Dispersion interaction was calculated by using the empirical parameterized Grimme (D3) method[41]. The valence-shell electrons ($1s^1$ for H, $2s^22p^2$ for C, $2s^22p^3$ for N, $3d^{10}4p^2$ for Zn, and $4s^24p^5$ for Br) are described using hybrid Gaussian and plane-wave (GPW) basis sets, cutoff energy of 800 Rydberg of auxiliary plane wave basis sets were adopted[42]. The excited states computations were performed using the the time-dependent density functional theory (TD-DFT) with the gaussian and augmented-plane-wave method[43]. It is well known that the GGA functionals severely underestimate the excitation energy but the hybrid usually provides much more accurate energy. It was reported that the PBE0 Hybrid functional gives a better description than the B3LYP functional for excitation energy calculation[44]. Therefore the Hybrid PBE0 was used to calculation the excited states[45]. The electron-hole analysis was performed using wavefunction analysis software of the Multiwfn[46]. The transition density matrix (TDM) of $(4DMAP)_2ZnBr_4$ was calculated under ambient conditions and upon complete pressure release using the Gaussian 16 software package[47]. To investigate the evolution of the crystal structure of $(4DMAP)_2ZnBr_4$ during compression and decompression, ab initio molecular dynamics (AIMD) simulations were conducted using VASP on a $2 \times 2 \times 1$ supercell comprising 360 atoms along the x and y axes. The NPT ensemble simulations were carried out at pressures of 0 GPa, 20 GPa, 30 GPa, 40 GPa, and 50 GPa for 5 ps with a time step of 1 fs and a temperature of 300 K. The microstructure when the system temperature is stable at 300 K is selected as the model after pressure is applied. The Brillouin zone was sampled during self-consistent iterations with k-point grids having a spacing of $2\pi \times 0.05/Å$, or better. The equilibrium structures were obtained through total energy minimization, with the residual forces on the atoms converged to below 0.005 eV/Å. Accounting for intermolecular interactions is important for the structural relaxation step. Standard DFT-D3 (IVDW = 12) is adopted. Geometry optimizations were performed to model the pressure release process, followed by calculations of radial distribution functions (RDF) and oscillator strengths[48].

## Steric Effect Index calculations

Steric effect index was calculated according to Eq. 2

$$SEI = \sum_{i=1}^{n} \frac{(R_{i,x}/R_C)^3}{(l_{i,x}/l_{C-C})^3} \quad (2)$$

where, $R_{i,x}$ is X (including C atom) atomic covalent radius, $R_C$ is C atomic covalent radius, $l_{i,x}$ is the sum of the bond length from atom X to the reference center. It is known that the carbon atomic, nitrogen covalent radius $R_C$ are 0.77, $0.74 \times 10^{-8}$ cm and the bond length $r_{C-C}$ of C-C, C-N are $(0.77 + 0.77)$, $(0.77 + 0.74) \times 10^{-8}$ cm, respectively. According to eq 3

$$SEI(4DMAP) = \frac{(R_C/R_C)^3}{(l_{C-N}/l_{C-C})^3} \times 2 + \frac{(R_C/R_C)^3}{(l_{C-N+C-C}/l_{C-C})^3} \times 2 + \frac{(R_C/R_C)^3}{(l_{C-N+C-C+C-C}/l_{C-C})^3} +$$

$$\frac{(R_N/R_C)^3}{(l_{C-N+C-C+C-C+C-N}/l_{C-C})^3} + \frac{(R_C/R_C)^3}{(l_{C-N+C-C+C-C+C-N+C-N}/l_{C-C})^3} \times 2 = 2.45$$

## Data availability

The authors declare that the main data supporting our findings of this study are contained within the paper and Supplementary Information. Source data are provided with this paper.

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

## Acknowledgements

This work is supported by the National Key R&D Program of China (2023YFA1406200 and 2022YFB3706602), the National Natural Science Foundation of China (12174144, 12474009, 12304014, 62304089 and 12304015), and the project of Interdisciplinary Subjects for Young Teachers and Students of Jilin University (2025-JCXK-09). The work at Linköping University was supported by the Knut and Alice Wallenberg Foundation (KAW 2019.0082). F.G. is a Wallenberg Scholar. This work was additionally supported by the Synergetic Extreme Condition User Facility (SECUF)-Jilin Branch, China. This work was mainly performed at BL15U1 at the Shanghai Synchrotron Radiation Facility (SSRF).

## Author contributions

G.X., L.Z., F.G., B.Z. designed the project and supervised the work. D.Z., S.L., Y.S., J.Q., G.X., Y.S., X.Y., P.L., F.W., J.Y., F.L., Q.Z., Z.L., L.Z., F.G. and B.Z. performed experiments and analyzed data. D.Z., Y.S. and X.Y. performed the calculations. All the authors participated in writing the manuscript and approved the version. D.Z., S.L., Y.S., J.Q. contributed equally to this work.

## Funding

## Competing interests
