## [Peer Review file · Nature Communications]

Pressure Encryption Toward Physically Uncopiable Anti-Counterfeiting

Corresponding Author: Professor Feng Gao

Version 0:

Reviewer comments:

Reviewer #1

(Remarks to the Author)

This paper reports pressure engineering dependence of retention of the enhanced multicolored emission upon different pressures releasing, which are attributed to controllable tuning of halogen-to-ligand charge transfer (HLCT) and local excitation (LE) implemented by pressure treatment. The abundant multicolored emissions exhibit great anti-counterfeiting potential. Pressure encryption is added in the current optical encryption using photoexcitation, which is fundamentally interesting and would highly strengthen the anti-counterfeiting security. Importantly, designated micro-nano patterns greatly enhance the security capability of current visual information encryption, which serves as the triple-level and physically uncopiable optical anti-counterfeiting technique. Overall, the data presented is reliable and supports the conclusions drawn with comprehensive discussions provided. Therefore, this manuscript is recommended to publish in Nat Commun after considering the following minor revisions.

- 1) As shown in Supplementary Figure 8, the single-point potential energy only offers a comparison of calculations between ambient pressure and high-pressure conditions. I am very curious about the comparison of calculations between ambient pressure and decompression conditions.
- 2) Authors claimed that a triple-level and physically uncopiable optical anti-counterfeiting was achieved by considering the quenched multicolored emissions after pressure treatment, which served as pressure engineering security key. However, the different emission intensity, especially the lifetime of those samples stabilized from different pressures are also an effective encryption information. Therefore, the emission intensities and lifetime of the multi-color emissions are suggested to be collected to enrich the manuscript.
- 3) The mentioned "Through loading different pressures, the initially non-emissive 0D organic metal hybrid halide (C₇H₁₁N₂, 4DMAP)₂ZnBr₄ shows at least 8 different distinct bright emission colors with varying emission intensities and lifetimes ...", the word "lifetimes" should not be in plural form. It is recommended that the author carefully review both the manuscript and supporting materials.
- 4) Reference formats should be identified, such as missing page numbers. Authors are suggested to thoroughly check that.

Reviewer #2

(Remarks to the Author)

In the manuscript, the authors present a multi-technique study on the pressure response of (4DMAP)₂ZnBr₄, which is a 0D organic metal halide hybrid material specifically designed and synthesized for the purposes of the work. (4DMAP)₂ZnBr₄ is a non-luminescent material at ambient conditions but exhibits emission under pressure, qualifying as a PIE (pressure-induced emission) luminogen. What makes this system particularly interesting is that

- i) the luminescence acquired under pressure is retained once the sample is brought back to ambient conditions;
- ii) by loading different pressures it is possible to obtain different bright luminescence colours in the visible range at ambient conditions.

The authors considered several samples, each one subjected to a pressure cycle up to a different maximum pressure,

ranging from 7 to 35 GPa. The optoelectronic response has been investigated by photoluminescence (PL) and absorption spectroscopies, while a combination of structural, vibrational and computational techniques is used to investigate the mechanisms underlying the sample behaviour.

Finally, the authors demonstrate a possible application of (4DMAP)₂ZnBr₄ as triple-level and physically uncopiable optical anti-counterfeiting materials.

The first level anti-counterfeiting mode is given by the possibility of selecting a unique luminescence wavelength for the samples; the second-level is given by the possibility of fabricating micro-nano patterns, due to the 0-D structure of the samples; the third-level is given by the regulation of the random distribution of the luminescent particles within the patterns. The synthesis of a brand-new crystal with remarkable optoelectronic properties, the multi-technique investigation of the physical mechanisms ruling the sample behaviour, and the applicability in the field of anti-counterfeiting technologies might make the manuscript of potential interest for a broad readership. However, there are several major points, criticisms, and comments that need to be addressed before really considering this article for publication.

Comments, criticisms, corrections in the following.

There completely lacks a description of the way high-pressure is applied. This is quite concerning given that the article is almost entirely focused on high-pressure measurements.

- I recommend including a description of the high-pressure setup used in each experiment (photoluminescence, infrared, absorption, etc.) in the Methods section. It seems that all the pressure measurements in the manuscript were performed by means of a diamond anvil cell (based also on the figures), while the samples used in the encryption process were compressed using a Walker-type large-volume press. This information must be explicit, and all the details concerning the two apparatuses should be given.

- For the measurements carried out with the Walker-type large-volume press it is necessary to explain the loading procedure.

- The choice of the pressure transmitting medium in each experiment should be discussed in detail. Indeed, this choice might influence the degree of hydrostaticity in the compression, which, in turn, might determine a different outcome in the material response.

There are cases in the literature in which the application of non-hydrostatic pressure produces irreversible structural transitions, while applying hydrostatic pressure on the same sample produces reversible transitions (e.g.

<https://doi.org/10.1063/1.4979143>, <https://doi.org/10.1103/PhysRevB.73.134101>).

In the present case, is it possible that the retainability of the optoelectronic response at ambient conditions is influenced by the degree of hydrostaticity of the compression?

- The kinetics of the compression should be elucidated, as it might play a fundamental role in the evolution of the sample properties and in the reversibility/irreversibility of the observed transitions.

In the manuscript, the authors perform pressure cycles up to a given maximum pressure PM to bring the sample from the initial non-emissive state to a final bright state, whose colour depends on PM.

How are these pressure cycles performed in each experiment? Is PM reached directly and continuously or step by step? What is the compression/decompression rate? After reaching PM, is the pressure immediately lowered, or a certain time interval is waited before decompressing?

Is it possible that a different kinetics in the pressure cycle determines a different response in the sample?

It remains unclear how the photoluminescence intensity varies in the samples R-7.1, ..., R-35.6 during and after loading different pressures.

The authors explain that the application of pressure tends to enhance the photoluminescence efficiency (as it is also visible from figure 3a in the supporting information). However, it is reasonable to assume that there exists a pressure threshold above which the intensity gain is interrupted. What is this threshold?

How does the intensity in the photoluminescence spectra of samples R-7.1, ..., R-35.6 (figure 1a) vary? Is there a correlation between the maximum pressure reached during the pressure cycle and the photoluminescence intensity retained at ambient conditions?

In figure 1, panels g and f are not particularly useful as they do not carry any quantitative information. A direct comparison between the spectrum at time 0 and after 800 h (or, equivalently between the spectrum after one cycle and after five cycles) would be much more significative to evaluate the reduction in intensity (see the last paragraph of the present list where the figures are all commented).

The results from IR and ADXRD should be evidenced more clearly. In the manuscript, the authors write:

Note that obvious redshift in the IR vibration and amorphous ADXRD pattern upon decompression are retained under ambient conditions, which coincides with retention of localized band-tail states (Supplementary Figs. 6b, 9, 10).

The redshift in the IR vibration is far from being obvious: it is extremely weak, and it is not well evidenced in the figures. The fitting procedure to extract the values for the peak centres and widths should be shown (at least in the supporting information).

Figures in the supporting information are poorly organized. Figure 6d and figure 9 show the IR vibrational modes of R12 and R21 respectively; figure 5d shows the IR vibrational modes on increasing pressure up to 21 GPa and after the pressure cycles R 1.9 and R4.9 GPa. Figure 3b in the main text shows a comparison between R21, R35 and the pristine sample. The authors cite figure 6b (why?) and 9, but they do not cite figures 5d and 3b. Everything is very confusing.

I recommend a unique figure in the main, in which the vibrational spectrum of the pristine sample (1 atm) is compared with all the series R-7.1, ..., R-35.6 in a way that makes the differences between the peaks clearly visible (for example the

spectra could be normalized).

The results from ADXRD should be moved from the supporting information to the main text. I would also suggest including in the figure the spectrum of more than one R- sample.

The formation of the aggregate states should be discussed in a cleared way.

The authors associate the presence of band-tail states in the absorption profile with the "aggregation of the compounds". What do they mean? Typically, the presence of a tail in the absorption profile is associated with the presence of disorder (Urbach tail). This point should be discussed in a more detailed way, possibly citing some relevant literature to support the interpretation of the data. Moreover, an unambiguous identification of this tail feature in the absorption spectra would be helpful.

How are these aggregates defined and what are their typical dimensions in the material? Does the amorphous nature of the aggregates play a role in the retainability of the photoluminescence signal at ambient conditions?

The mechanism of pressure-induced suppression of the molecular rotation could be compared with relevant literature on hybrid (organic-inorganic) compounds (e.g. doi: 10.1021/acs.jpcc.7b11461J , doi: 10.1021/acs.jpcc.2c08253). In HOIPs (hybrid organic inorganic perovskites), for example, the application of pressure drives a transition from a dynamically disordered phase at ambient conditions, where the molecules freely rotate, to a statically disordered phase at high pressure. In the latter case, the molecules are locked at random orientations giving rise to an amorphous-like phase. Remarkably, in HOIPs, this amorphous-like phase is often reversible, meaning that the original crystalline configuration is restored once the sample is brought back at ambient conditions.

Does a similar mechanism occur in (4DMAP)₂ZnBr₄? Do the authors have an idea of why, in this case, the transformation is irreversible?

Could the authors provide a concise explanation of what are Hirshfeld calculations? The figures (2h, 2f) reported in the text are obscure for a reader who is not a specialist in the field. What do they describe? What can we learn from them?

Could the author explain in detail how the images reported in figure 3 are obtained?

What are the substrates used? How is the material deposited on these substrates? How are the patterns created? Is pressure applied before or after the deposition on the substrate? If pressure is applied after the deposition of the material on the substrate, how is the press loaded?

The authors should provide much more detail on this procedure and, possibly, also some images of the sample preparation and loading.

The introduction section lacks a thoughtful review of the state of the art in the field of PIE luminogen.

Some relevant aspects, such as the retainability of the photoluminescence signal at ambient conditions and the wavelength range typically spanned, should be discussed.

Moreover, in the short paragraph devoted to the discussion of PIE, I found that some of the cited articles seem to be out of topic:

- [16] Modulating Charge-Density Wave Order and Superconductivity from Two Alternative Stacked Monolayers in a Bulk 4Hb-TaSe₂ Heterostructure via Pressure.
- [17] Pressure-Driven Reversible Switching between n- and p-Type Conduction in Chalcopyrite CuFeS₂
- [18] A Protocol to Fabricate Nanostructured New Phase: B31-Type MnS Synthesized under High Pressure
- [27] Ultrahard bulk amorphous carbon from collapsed fullerene
- [28] Identification of Defect Origin and White-light Emission Tuning of Chalcogenide Quantum Dots through Pressure Engineering.

I suggest replacing these articles with more appropriate references or explain better the relevance with the present article.

I would recommend a reorganization of the final section of the manuscript, as some repetitions are present in the text.

"Additionally, the fluorescent patterns can carry encrypted information, such as microscopic QR code labels. These labels can be deciphered using microscopic fluorescence imaging, enabling information retrieval from the embedded website information."

[...]

"Additionally, the fluorescent patterns of cyan-blue emission could also carry encrypted information, such as microscopic QR code labels. These labels can be deciphered using microscopic fluorescence imaging, enabling information retrieval from the embedded website information."

I would recommend a reorganization of the figures."

Most of the panels in figure 1 are redundant:

- The information contained in panel c is basically the same as in a and b.
- Panel d is identical to panel 6a.
- Panel e is unnecessary, the same information can be given in the text without a dedicated figure.
- Comments about panels f,g can be found above.

- A panel showing the intensity values for the photoluminescence signal of all the samples R-7.1, ..., R-35.6 would be particularly useful.

A figure is missing which reports the results from ADXRD in the main text.

Reviewer #3

(Remarks to the Author)

Reviewer #4

(Remarks to the Author)

Comments:

This manuscript propose a triple-level and physically uncopyable optical anti-counterfeiting modal 0D organic metal hybrid halides by pressure treatment. This is an interesting topic about of halides perovskites. Through detailed experiment measurement and theoretical calculations, the authors report pressure engineering dependence of controllable multicolored emissions upon different pressures releasing, which are attributed to controllable tuning of halogen-to-ligand charge transfer and local excitation implemented by pressure treatment. The manuscript is well written. The results of experiment optical characterization and theory calculations are reasonable. I haven't found errors or inconsistencies between them. I would like to suggest the publication of this manuscript in Nat Commun, however, some comments are given below for the further revisions.

Comment 1: In this study, molecular dynamics simulations are employed to illustrate the microstructure under varying pressures. The temperature of the system typically rises when subjected to pressure. Thus, it is crucial to include temperature change data throughout the process of molecular dynamics simulation in order to confirm that the resulting structure corresponds to 300K, thus enhancing consistency with experimental results.

Comment 2: Upon decompression, geometric optimization was applied in the calculations to obtain the amorphous microstructures. Based on what basis or reference were the calculation method chosen?

Comment 3: Supplementary Figure S12 illustrates the microstructures following different pressure release conditions. It is essential to label these microstructures as subfigures, maintaining consistency with the labeling format utilized in other Figures.

Comment 4: The molecular dynamics simulation calculation method should be included in the section outlining the calculations in the main text rather than being relegated to supplementary materials.

Comment 5: Pressure induces changes in the types of excited states, a phenomenon of great interest. However, only partial information regarding these states is presented in Figure 2 and Supplementary Figure S8. It is suggested that the authors determine and report on specific excited states along with their corresponding types to confirm the accuracy of the provided content.

Comment 6: As illustrated in Figure 2c, what is the reason for the insignificantly low calculated oscillator strength intensity, and which method was employed?

Version 1:

Reviewer comments:

Reviewer #1

(Remarks to the Author)

The revised paper meets the publication requirement now, accept.

Reviewer #2

(Remarks to the Author)

The authors have done hard and serious work, and we believe that the paper has been remarkably improved. We are now confident that the present version of the manuscript deserves publication on Nature Communication without any further revision.

Reviewer #3

(Remarks to the Author)

Reviewer #4

(Remarks to the Author)

I examined the revised manuscript and compared the changes made by the author in response to my report. The authors have well answered my questions. I am satisfied with the paper. The current manuscript can be published as is.

Response to Reviewer #1

Comments: This paper reports pressure engineering dependence of retention of the enhanced multicolored emission upon different pressures releasing, which are attributed to controllable tuning of halogen-to-ligand charge transfer (HLCT) and local excitation (LE) implemented by pressure treatment. The abundant multicolored emissions exhibit great anti-counterfeiting potential. Pressure encryption is added in the current optical encryption using photoexcitation, which is fundamentally interesting and would highly strengthen the anti-counterfeiting security. Importantly, designated micro-nano patterns greatly enhance the security capability of current visual information encryption, which serves as the triple-level and physically uncopiable optical anti-counterfeiting technique. Overall, the data presented is reliable and supports the conclusions drawn with comprehensive discussions provided. Therefore, this manuscript is recommended to publish in Nat Commun after considering the following minor revisions.

1) As shown in Supplementary Figure 8, the single-point potential energy only offers a comparison of calculations between ambient pressure and high-pressure conditions. I am very curious about the comparison of calculations between ambient pressure and decompression conditions.

Author reply: We appreciate the insightful comments and constructive suggestions. In response to the reviewer's comments, we have included the comparison of calculations for the single-point potential energy between the ambient conditions and the decompression conditions in Supplementary Fig. 14 and Supplementary Table II of the revised manuscript. The single-point potential energy was calculated using DFTB⁺ with dispersion correction. Here, the calculated rotational single-point energy represents that the energy required for 4DMAP⁺ rotation without hindrance is increased significantly under high pressure, and this effect persists even after the pressure is released completely.

The corresponding revised details are highlighted in yellow and can be found in **Supplementary Fig. 14 and Table SII of the Supplementary Materials in the revised manuscript.**

Added Supplementary Fig. 14. Calculated Single-point energy for the rotation of DMAP⁺ rings in (4DMAP)₂ZnBr₄. Single-point energy at 1 atm (blue) and at R-20 GPa decompressed from 20 GPa (red), respectively.

Added Supplementary Table II. Calculated single-point energy for the rotation of 4DMAP⁺ rings in (4DMAP)₂ZnBr₄ using DFTB+ calculations for 0 GPa and R-20 GPa.

Rotation angle (°)	0 GPa (kcal/mol)	R-20 (kcal/mol)
0	-72959.18	-72929.23
10	-72956.08	-72920.14
20	-72947.26	-72893.11
30	-72931.68	-72843.81
40	-72903.61	-72764.74
50	-72860.54	-72661.08
60	-72803.84	-72554.83
70	-72735.61	-72464.27
80	-72647.79	-72420.08
90	-72652.96	-72444.17
100	-72658.26	-72502.71
110	-72693.38	-72593.36
120	-72750.44	-72692.69
130	-72817.09	-72771.01
140	-72873.56	-72828.01
150	-72914.96	-72874.16
160	-72940.99	-72901.51
170	-72954.17	-72918.83
180	-72958.35	-72926.82
190	-72955.61	-72919.64
200	-72946.98	-72893.53
210	-72931.46	-72846.37
220	-72903.51	-72771.62
230	-72862.82	-72667.93
240	-72805.91	-72565.26
250	-72733.28	-72473.85
260	-72666.31	-72427.84
270	-72640.49	-72447.59
280	-72647.98	-72502.82
290	-72686.41	-72506.54
300	-72747.12	-72694.55
310	-72815.73	-72773.45
320	-72873.91	-72828.15
330	-72915.66	-72874.29
340	-72941.78	-72904.44
350	-72955.11	-72921.95
360	-72959.18	-72928.94

2) Authors claimed that a triple-level and physically uncopyable optical anti-counterfeiting was achieved by considering the quenched multicolored emissions after pressure treatment, which served as pressure engineering security key. However, the different emission intensity, especially the lifetime of those samples stabilized from different pressures are also an effective encryption information. Therefore, the emission intensities and lifetime of the multi-color emissions are suggested to be collected to enrich the manuscript.

Author reply: We appreciate the constructive suggestions. We have included the emission intensities (Fig. 1c) and PL lifetime (Supplementary Fig. 4) of the multi-color emissions in the revised manuscript. Here, we should point out that it is the first time we reveal bright multi-color tunability through loading pressure and releasing it, rather than the conventional ways by changing material component, tuning excitation wavelength or introducing heating. Furthermore, variations in PL intensity and lifetime after loading different pressures contributed to the increased multidimensionality of the pressure engineering secret key (PESK). Thus, we provide a new strategy with high encryption level for the PESK application from PIEgens (Fig. 1c and Supplementary Fig. 4).

Revised Fig. 1. Potential of (4DMAP)₂ZnBr₄ as the PESK application. c Intensity values for the photoluminescence signal of all the samples R-7.1, ..., R-35.6.

Added Supplementary Fig. 4. PL decay curves after loading different pressures. **a** and **b** Changes in PL decay of high-energy and low-energy pinks of $(4\text{DMAP})_2\text{ZnBr}_4$ after loading different pressures, respectively. **c** Lifetime varies in the samples R-7.1, ..., 35.6 after loading different pressures.

The corresponding revised details are highlighted in yellow and can be found in Line 6-8, Page 5 of the Manuscript and Supplementary Fig. 4 of the Supplementary Materials in the revised manuscript.

3) The mentioned “Through loading different pressures, the initially non-emissive 0D organic metal hybrid halide $(\text{C}_7\text{H}_{11}\text{N}_2, 4\text{DMAP})_2\text{ZnBr}_4$ shows at least 8 different distinct bright emission colors with varying emission intensities and lifetimes ...”, the word “lifetimes” should not be in plural form. It is recommended that the author carefully review both the manuscript and supporting materials.

Author reply: Following the reviewer's recommendations, we have thoroughly reviewed both the Manuscript and the Supplementary Materials and carefully revised the word “lifetimes” to “lifetime” in both the Manuscript and the Supplementary Materials in the revised manuscript.

The corresponding revised details are highlighted in yellow and can be found in full text of the Manuscript and the Supplementary Materials in the revised manuscript.

4) References formats should be identified, such as missing page numbers. Authors are suggested to thoroughly check that.

Author reply: Following the reviewer's kind suggestions, we have thoroughly reviewed both the Manuscript and the Supplementary Materials for references formats and carefully revised the references formats in both the Manuscript and Supplementary Materials in the revised manuscript.

The corresponding revised details are highlighted in yellow and can be found in the Manuscript and the Supplementary Materials in the revised manuscript.

Response to Reviewer #2

Comments: In the manuscript, the authors present a multi-technique study on the pressure response of $(4\text{DMAP})_2\text{ZnBr}_4$, which is a 0D organic metal halide hybrid material specifically designed and synthesized for the purposes of the work. $(4\text{DMAP})_2\text{ZnBr}_4$ is non-luminescent material at ambient conditions but exhibits emission under pressure, qualifying as a PIE (pressure-induced emission) luminogen. What makes this system particularly interesting is that

i) the luminescence acquired under pressure is retained once the sample is brought back to ambient conditions;

ii) by loading different pressures, it is possible to obtain different bright luminescence colours in the visible range at ambient conditions.

The authors considered several samples, each one subjected to a pressure cycle up to a different maximum pressure, ranging from 7 to 35 GPa. The optoelectronic response has been investigated by photoluminescence (PL) and absorption spectroscopies, while a combination of structural, vibrational and computational techniques is used to investigate the mechanisms underlying the sample behavior.

Finally, the authors demonstrate a possible application of $(4\text{DMAP})_2\text{ZnBr}_4$ as triple-level and physically uncopiable optical anti-counterfeiting materials. The first level anti-counterfeiting mode is given by the possibility of selecting a unique luminescence wavelength for the samples; the second-level is given by the possibility of fabricating micro-nano patterns, due to the 0-D structure of the samples; the third-level is given by the regulation of the random distribution of the luminescent particles within the patterns. The synthesis of a brand-new crystal with remarkable optoelectronic properties, the multi-technique investigation of the physical mechanisms ruling the sample behavior, and the applicability in the field of anti-counterfeiting technologies might make the manuscript of potential interest for a broad readership. However, there are several major points, criticisms, and comments that need to be addressed before really considering this article for publication.

Comments 1: There completely lacks a description of the way high-pressure is applied. This is quite concerning given that the article is almost entirely focused on high-pressure measurements. I recommend including a description of the high-pressure setup used in each experiment (photoluminescence, infrared, absorption, etc.) in the Methods section. It seems that all the pressure measurements in the manuscript were performed by means of a diamond anvil cell (based also on the figures), while the samples used in the encryption process were compressed using a Walker-type large-

volume press. This information must be explicit, and all the details concerning the two apparatuses should be given.

Author reply: Following the reviewer's suggestions, we have included a description of the high-pressure setup used in each experiment (photoluminescence, infrared, absorption, etc.) and Walker-type large-volume press in the Methods section of the revised manuscript. In our work, the steps are as follows: Firstly, we pre-pressed the thickness of the T301 steel plate to about 45 micrometers using a diamond anvil cell (DAC). Then we used a laser to drill a hole with a diameter of one-third of T301 steel plate, where the hole is used as the sample chamber. In the PL experiment, the powder samples are uniformly dispersed in silicone oil, which acts as a pressure-transmitting medium to establish a hydrostatic environment. Similarly, in the UV-vis absorption experiment, the suppressed transparent bulk thin layer sample is immersed in silicone oil, creating a hydrostatic pressure environment around the sample. At the same time, Ruby is used for pressure calibration inside the hole. It should be noted that during the infrared experiment, due to the peaks associated with silicone oil, KBr is used as the pressure transmitting medium. All experiments were conducted under room temperature conditions.

In response to the reviewer's inquiry regarding the use of a Walker-type large-volume press for the encryption process, we have provided a detailed explanation. The availability of samples prepared using a DAC is extremely limited, which is insufficient to meet the demands for producing a variety of anti-counterfeiting and information storage patterns. As a result, the preparation of large mass samples necessitates the use of a Walker-type large-volume press. Here, the DAC is employed to investigate pressure-induced changes in physical properties, while the Walker-type large-volume press is utilized to prepare large mass samples. The combination of a diamond anvil cell and a Walker-type large-volume press has proven highly effective in advancing the practical application of metastable optoelectronic functional materials. For instance, this approach has been successfully applied in the fabrication of phosphor-converted light-emitting diodes (pc-LEDs).^{1,2}

References

1. Xiao, Z. et al. Harvesting Multicolor Photoluminescence in Nonaromatic Interpenetrated Metal–Organic Framework Nanocrystals via Pressure-Modulated Carbonyls Aggregation. *Adv. Mater.* 36, 2403281 (2024).
2. Yang, Q. et al. Pressure treatment enables white-light emission in Zn-IPA MOF via asymmetrical metal-ligand chelate coordination. *Nat. Commun.* 16, 696 (2025).

The corresponding revised details are highlighted in yellow and can be found in **Line 3-10, Page 16 of the Manuscript in the revised manuscript.**

Comments 2: *For the measurements carried out with the Walker-type large-volume press it is necessary to explain the loading procedure.*

Author reply: In response to the reviewer's suggestion, we have included a description of the measurements carried out with the Walker-type large-volume press in the Methods section of the revised manuscript. The details are as follows: High-pressure quench experiments were conducted at 12 GPa and 21 GPa by using a 10/5 (OEL/TEL = octahedral edge length of pressure medium/truncated edge length of anvil) cell assembly in a 10-MN Walker-type large-volume press at the State Key Laboratory of High Pressure and Superhard Materials, Jilin University.¹ The starting (4DMAP)₂ZnBr₄ powders were housed in a rhenium capsule and placed into MgO₂ sleeves within a Cr₂O₃-doped MgO octahedron. Pressure calibration of the cell assembly at room temperature had been previously reported.¹ Initially, the samples were compressed to predetermined pressures over a span of approximately 10 h at room temperature, followed by a gradual pressure release lasting roughly 10 h.

References

1 Shang, Y. C. et al. Pressure Generation above 35 GPa in a Walker-Type Large-Volume Press. *Chin. Phys. Lett.* 37, 080701 (2020).

The corresponding revised details are highlighted in yellow and can be found in Line 18-26, Page 16 of the Manuscript in the revised manuscript.

Comments 3: *The choice of the pressure transmitting medium in each experiment should be discussed in detail. Indeed, this choice might influence the degree of hydrostaticity in the compression, which, in turn, might determine a different outcome in the material response. There are cases in the literature in which the application of non-hydrostatic pressure produces irreversible structural transitions, while applying hydrostatic pressure on the same sample produces reversible transitions (e.g. <https://doi.org/10.1063/1.4979143>, <https://doi.org/10.1103/PhysRevB.73.134101>). In the present case, is it possible that the retainability of the optoelectronic response at ambient conditions is influenced by the degree of hydrostaticity of the compression?*

Author reply: We thank the reviewer for this comment. In our study, in the PL experiment, the powder samples are uniformly dispersed in silicone oil, which acts as a pressure-transmitting medium to establish a hydrostatic environment. Similarly, in the UV-vis absorption experiment, the suppressed transparent bulk thin layer sample is immersed in silicone oil, creating a hydrostatic pressure environment around the sample. At the same time, Ruby is used for pressure calibration inside the hole. It should be noted that during the infrared experiment, due to the peaks associated with silicone

oil, KBr is used as the pressure transmitting medium. All experiments were conducted under room temperature conditions.

Indeed, the choice of the pressure transmitting medium might influence the degree of hydrostaticity in the compression, which, in turn, might determine a different outcome in the material response. In the case of the report (<https://doi.org/10.1063/1.4979143>), the degree of hydrostaticity of the compression has a significant impact on the retainability of the optoelectronic response at ambient conditions, where the metallization becomes irreversible only under non-hydrostatic compression. A reasonable explanation for the different properties displayed by MoS₂ under non-hydrostatic and hydrostatic conditions is the influence of pressure medium and anisotropic stress. For hydrostatic conditions, the molecules of methanol and ethanol can enter the interlayer space to alleviate the interlayer interactions. After decompression, the layer spacing recovers with the escape of methanol and ethanol molecules. For the non-hydrostatic sample, a stronger interaction is generated in the van der Waals gap, preventing the recovery of the interlayer spacing after compression owing to the absence of methyl alcohol molecules and the presence of strong anisotropic stress. Meanwhile, in another case (<https://doi.org/10.1103/PhysRevB.73.134101>), usually at pressures up to 100 kbar (10 GPa), the pressure transmitting medium yields nearly hydrostatic conditions.

In order to investigate the influence of the pressure transmitting medium on the degree of hydrostaticity during compression, the choice of pressure transmitting medium for each experiment is discussed in detail. Based on the literature provided by the reviewer, we conducted the relevant experiments using silicone oil and ethanol/methanol with a volume ratio of 4:1 as the pressure transmitting medium to systematically examine the influence of the medium on the degree of hydrostaticity. The experiments were performed on samples R-5.2, R-4.1, R-3.1, R-2.1, R-1.1, and 0, as depicted in added Supplementary Fig. 8a and 8b. Upon comparing the PL spectra from the two scenarios, it is evident that pressure-induced emission is present in both cases, with no discernible difference in the spectra between them. Furthermore, we conducted a comparison of photoluminescence after loading high pressures up to 21.5 GPa. The results revealed minimal differences, as shown in added Supplementary Fig. 8c. Unlike ABO₃-type perovskite structures, (4DMAP)₂ZnBr₄, as low-dimensional hybrid halides, do not strictly adhere to the Goldschmidt tolerance factor rules.¹ The incorporation of large condensed ring amine cations 4DMAP⁺ at the A sites is permitted. The limited space facilitates the formation of robust hydrogen bonds both among the cations and between these cations and the Br⁻ ions. Therefore, the (4DMAP)₂ZnBr₄ material exhibits a highly dense structure. As a result, the molecules of the pressure transmitting medium are unable to penetrate the material, merely providing a hydrostatic pressure environment on its surface,

leaving these interactions unaffected. Therefore, the selection of the pressure transmitting medium does not significantly affect our experimental findings.

Added Supplementary Fig. 8. Influence of the degree of hydrostaticity of the compression on the retainability of the optoelectronic response at ambient conditions. a and b Silicone oil and ethanol/methanol with a volume ratio of 4:1 used to be a pressure transmitting medium to investigate inflections of the degree of hydrostaticity of the compression. c Comparison of PL spectra at R-21.5 GPa using silicone oil and ethanol/methanol as the pressure transmitting medium.

References

1 Li, X., Hoffman, J. M. & Kanatzidis, M. G. The 2D halide perovskite rulebook: how the spacer influences everything from the structure to optoelectronic device efficiency. *Chem. Rev.* 121, 2230-2291 (2021).

The corresponding revised details are highlighted in yellow and can be found in Line 12-29, Page 6 of the Manuscript and Supplementary Fig. 8 of the Supplementary Materials in the revised manuscript.

Comments 4: The kinetics of the compression should be elucidated, as it might play a fundamental role in the evolution of the sample properties and in the reversibility/irreversibility of the observed transitions.

In the manuscript, the authors perform pressure cycles up to a given maximum pressure PM to bring the sample from the initial non-emissive state to a final bright state, whose color depends on PM . How are these pressure cycles performed in each experiment? Is PM reached directly and continuously or

step by step? What is the compression/decompression rate? After reaching PM, is the pressure immediately lowered, or a certain time interval is waited before decompressing? Is it possible that a different kinetics in the pressure cycle determines a different response in the sample?

Author reply: Following the comment from the reviewer, we have performed additional high-pressure experiments. In the pressure cycles of our work, the compression and decompression processes were carried out step by step gradually to maintain static high-pressure equilibrium. The maximum pressure PM was reached step by step. Throughout this process, the compression and decompression rates were set at 1 GPa/5min, respectively. After reaching PM, the pressure is immediately lowered step by step. To investigate different kinetics during the pressure cycle and determine a different response in the sample, we conducted experiments with different compression rates, including 1 GPa/5min, 1 GPa/3min, 1 GPa/1min, 1 GPa/0.5min, as shown in added Supplementary Fig. 9 of the revised manuscript. The compression rate below 1 GPa/1min has a negligible effect on the emissions. However, when the compression rate exceeds 1 GPa/0.5min, there will be a decrease in emission intensity, attributed to the enhanced amorphous degree.¹ Therefore, as pointed out by the reviewer, it is possible that different kinetics in the pressure cycle determines a different response in the sample.

Added Supplementary Fig. 9. Different kinetics in the pressure cycle R-21.5 GPa in response to the PL changes, including 1 GPa/5min, 1 GPa/3min, 1 GPa/1min, 1 GPa/0.5min.

References

1 Shi, K. et al. Sulfur chains glass formed by fast compression. *Nat. Commun.* 16, 357 (2025).

The corresponding revised details are highlighted in yellow and can be found in Line 31-34, 9-10, Page 6, 7 of the Manuscript and Supplementary Fig. 9 of the Supplementary Materials in the revised manuscript.

Comments 5: It remains unclear how the photoluminescence intensity varies in the samples R-7.1, ..., R-35.6 during and after loading different pressures.

The authors explain that the application of pressure tends to enhance the photoluminescence efficiency (as it is also visible from figure 3a in the supporting information). However, it is reasonable to assume that there exists a pressure threshold above which the intensity gain is interrupted. What is this threshold? How does the intensity in the photoluminescence spectra of samples R-7.1, ..., R-35.6 (figure 1a) vary? Is there a correlation between the maximum pressure reached during the pressure cycle and the photoluminescence intensity retained at ambient conditions? In figure 1, panels g and f are not particularly useful as they do not carry any quantitative information. A direct comparison between the spectrum at time 0 and after 800 h (or, equivalently between the spectrum after one cycle and after five cycles) would be much more significative to evaluate the reduction in intensity (see the last paragraph of the present list where the figures are all commented).

Author reply: Following the reviewer's comment, we have added supporting information. As shown in revised Supplementary Fig. 2a-2c, during loading pressures, the intensity of the emission initially increases from 1 atm to the pressure threshold of 7.2 GPa, followed by a decrease from 7.2 GPa to 21.5 GPa. In addition, after loading different pressures, the emission intensity first rises from R-7.1 to R-21.5 GPa and subsequently declines from R-21.5 to R-35.6 GPa (revised Supplementary Fig. 2d and 2e).

Furthermore, we performed additional high-pressure experiments to investigate the threshold of pressure treatment to achieve the emission from the initial non-emissive $(4\text{DMAP})_2\text{ZnBr}_4$. As shown in added Supplementary Fig. 8a and 8b, after conducting several experiments under low pressures, no obvious emission can be observed after pressure treatment when the applied pressure is below 2 GPa. Therefore, just as the reviewer expected, we have determined 2 GPa as the threshold of pressure treatment for the retention of PIE.

Moreover, there is a correlation between the maximum pressure reached during the pressure cycle and the photoluminescence intensity retained at ambient conditions. As illustrated in the added Supplementary Fig. 2e, the photoluminescence intensity retained initially increases and subsequently decreases with the application of increasing pressure cycle treatment. In addition, the PL emission

spectra experienced a slight decrease in intensity after experiencing five cycles, indicating good stability of the pressure cycle (added Supplementary Fig. 7).

In response to the reviewer's suggestions, we have revised the Fig. 1. Fig. 1 demonstrates, as anticipated by reviewer, a direct comparison between the PL spectra at 0 and after 800 h (revised Fig. 1d), as well as between the PL spectra after one cycle and after five cycles (revised Fig. 1e). This comparison proves to be more significant to evaluate the reduction in intensity.

Revised Supplementary Fig. 2. PL spectra and intensity varies in the samples during and after loading different pressures. a and b Changes in PL spectra of $(4\text{DMAP})_2\text{ZnBr}_4$ under high pressure. **c** PL intensity varies in the samples 1 atm, ..., 21.5 during loading different pressures. **d** Changes in PL spectra after loading different pressures. **e** PL intensity varies in the samples R-7.1, ..., R-35.6 after loading different pressures. **f** Comparison of PL spectra at 1 atm and R-21.5 GPa.

Added Supplementary Fig. 8. Influence of the degree of hydrostaticity of the compression on the retainability of the optoelectronic response at ambient conditions.

Added Supplementary Fig. 7. Integrated PL intensity of quenched $(4\text{DMAP})_2\text{ZnBr}_4$ as a function of cycle numbers by releasing from pressure of 21.5 GPa. Inset displays the corresponding PL intensity changes.

Revised Fig. 1. Potential of $(4\text{DMAP})_2\text{ZnBr}_4$ as the PESK application. **a** Normalized PL spectra of $(4\text{DMAP})_2\text{ZnBr}_4$ measured at 1 atm after loading different pressures. **b** Relationship between emission energy and released pressure from different pressure engineering points. **c** Intensity values for the photoluminescence signal of all the samples R-7.1, ..., R-35.6. **d** and **e** PL intensity of quenched $(4\text{DMAP})_2\text{ZnBr}_4$ as a function of aging time and cycle numbers by releasing from pressure of 21.5 GPa, respectively.

The corresponding revised details are highlighted in yellow and can be found in Line 30-31, Page 6, Fig. 1 of the Manuscript and Supplementary Figs. 2, 7, 8 of the Supplementary Materials in the revised manuscript.

Comments 6: *The results from IR and ADXRD should be evidenced more clearly. In the manuscript, the authors write: Note that obvious redshift in the IR vibration and amorphous ADXRD pattern upon decompression are retained under ambient conditions, which coincides with retention of localized band-tail states (Supplementary Figs. 6b, 9, 10). The redshift in the IR vibration is far from being obvious: it is extremely weak, and it is not well evidenced in the figures. The fitting procedure to extract the values for the peak centres and widths should be shown (at least in the supporting information). Figures in the supporting information are poorly organized. Figure 6d and figure 9 show the IR vibrational modes of R12 and R21 respectively; figure 5d shows the IR vibrational modes on increasing pressure up to 21 GPa and after the pressure cycles R 1.9 and R4.9 GPa. Figure 3b in the main text shows a comparison between R21, R35 and the pristine sample. The authors cite figure 6b (why?) and 9, but they do not cite figures 5d and 3b. Everything is very confusing. I recommend a unique figure in the main, in which the vibrational spectrum of the pristine sample (1 atm) is compared with all the series R-7.1, ..., R-35.6 in a way that makes the differences between the peaks clearly visible (for example the spectra could be normalized). The results from ADXRD should be moved from the supporting information to the main text. I would also suggest including in the figure the spectrum of more than one R- sample.*

Author reply: We highly appreciate the reviewer for the insightful comments. Given the low strength of the infrared signal, we performed new IR experiments with an extended integration time to improve data quality. Following the reviewer's constructive suggestions, we have included the fitting procedure to extract peak centers and full width at half maximum (FWHM) values in the newly added Supplementary Fig. 11 and Fig. 3c, respectively. This result clearly illustrates the characteristics of redshift in the IR vibration. Recognizing the suboptimal organization of figures in the supporting information, we have thoroughly reorganized these materials. In response to the reviewer's valuable suggestions, in the revised manuscript, we have also added Fig. 2 of the results from ADXRD and Fig. 3 of IR vibration to the main text to better present these findings.

Added Supplementary Fig. 11. Pink centers of IR vibrational spectrum. a-f Fitting procedure to extract the values for the IR peak centers using Gaussian after decompression from different pressures.

Added Fig. 3. Changes in vibrational spectrum during and after loading different pressures. **a** Evolution of vibrational spectrum of $(4\text{DMAP})_2\text{ZnBr}_4$ under high pressure. **b** Evolution of vibrational spectrum of $(4\text{DMAP})_2\text{ZnBr}_4$ after decompression. **c** Intensity and FWHM evolution of vibrational spectrum of $(4\text{DMAP})_2\text{ZnBr}_4$ after decompression.

Added Fig. 2. Changes in ADXRD and lattice parameters during and after loading different pressures. **a** Powder X-ray diffraction patterns representative of $(4\text{DMAP})_2\text{ZnBr}_4$ under high pressure. **b** Experimental lattice constants (a , b and c) and volume of $(4\text{DMAP})_2\text{ZnBr}_4$ as a function of pressures. The solid lines represent the Birch-Murnaghan EOS functions fitted to the measured P-V data. **c** Comparison of refinement results for $(4\text{DMAP})_2\text{ZnBr}_4$ at 1 atm, R-21.5 and R-12.2 GPa, respectively. The purple bars indicate the positions of the refined Bragg peaks, while the black line represents the difference between the experimental (red crosswire) and simulated calculated (black solid line) diffraction profiles.

The corresponding revised details are highlighted in yellow and can be found in Figs. 2, 3 of the Manuscript and Supplementary Fig. 11 of the Supplementary Materials in the revised manuscript.

Comments 7: *The formation of the aggregate states should be discussed in a cleared way.*

The authors associate the presence of band-tail states in the absorption profile with the “aggregation of the compounds”. What do they mean? Typically, the presence of a tail in the absorption profile is associated with the presence of disorder (Urbach tail). This point should be discussed in a more detailed way, possibly citing some relevant literature to support the interpretation of the data. Moreover, an unambiguous identification of this tail feature in the absorption spectra would be helpful. How are these aggregates defined and what are their typical dimensions in the material? Does the amorphous nature of the aggregates play a role in the retainability of the photoluminescence signal at ambient conditions?

Author reply: We are grateful to the reviewer for the insightful comments and constructive suggestions. In our study, an amorphous phase was observed to form under high-pressure conditions (see following Figure 1a), leading to the emergence of a novel electronic state associated with the development of amorphous disorder in the aggregated compounds (Figure 1b). As a result, the newly formed electronic structure is intricately linked to the amorphous aggregate state. Upon decompression, the residual new electronic state in the quenched amorphous products indicates that the quenched structure retains its disordered nature (Figure 1c). Regarding the identification of this new electronic state, we further added some discussions. For instance, in amorphous silicon (α -Si), this amorphous nature leads to localized states in the band tails, known as the Urbach tail. To enhance the precise identification of the tail feature in the absorption spectra, we have revised the term "band-tail states" to "Urbach tail" after carefully considering the valuable suggestions provided by the reviewer.¹⁻³ Meanwhile, we have already revised the relevant descriptions in the full text. In addition, we have cited some relevant literatures to support the interpretation of the Urbach tail.¹⁻³ Once again, we express our gratitude for the reviewer's thoughtful suggestions, which have significantly improved the quality of our work.

In response to the reviewer's inquiry regarding the definition of these aggregates and their typical dimensions in the material, the explanation is as follows. The mechanism underlying the Aggregation-Induced Emission (AIE) phenomenon, as proposed by Professor Benzhong Tang, primarily involves the restriction of intramolecular motion and the enhancement of intermolecular interactions.¹ In our (4DMAP)₂ZnBr₄ system, hybrid halides fail to strictly adhere to the Goldschmidt tolerance factor rules, allowing for the incorporation of large condensed-ring amine cations 4DMAP⁺ at the A sites, whose limited space results in the robust hydrogen bonds formed both among cations and between these cations and Br⁻ ions. After high-pressure engineering treatment, enhanced hydrogen bonding restricts intramolecular motion and strengthens intermolecular interactions. Therefore, the state achieved after

high-pressure engineering treatment can be described as an aggregated state analogous to that observed in AIE phenomenon.

The amorphous aggregate state plays a crucial role in the retainability of the photoluminescence signal at ambient conditions, such as self-stabilized amorphous organic materials with room-temperature phosphorescence.⁴ Here, the amorphous aggregate state facilitates the strengthening of hydrogen bonding interactions both among the condensed ring amine cations 4DMAP⁺ and between these cations and inorganic Br⁻ component (see following Figure 2a and 2b). This results in the formation of molecular aggregates with increased steric hindrance effects, thereby raising the phase transition barrier (Figure 2d). Consequently, the metastable phase is stabilized, ensuring the retention of a high-quality photoluminescence signal under ambient conditions. The irreversible absorption behavior of organic molecules DMAP (Figure 2c) and the strengthened hydrogen bonding after pressure treatment further confirmed this idea.

References

1. Pan, Y., Inam, F., Zhang, M. & Drabold, D. A. Atomistic Origin of Urbach Tails in Amorphous Silicon. *Phys. Rev. Lett.* **100**, 206403 (2008).
2. Luo, Y. & Flewitt, A. J. Understanding localized states in the band tails of amorphous semiconductors exemplified by a-Si: H from the perspective of excess delocalized charges. *Phys. Rev. B* **109**, 104203 (2024).
3. Rambadey, O. V., Kumar, K. & Sagdeo, P. R. Temperature dependence of disorder sensitivity of phonon modes in finite-gap materials. *Phys. Rev. B* **110**, 235201 (2024).
- 4 Xu, W. et al. Self-Stabilized Amorphous Organic Materials with Room-Temperature Phosphorescence. *Angew. Chem. Int. Ed.* **58**, 16018-16022 (2019).

Figure 1. **a** Changes in ADXRD during and after loading different pressures. **b** Characterization of optical absorption under high pressure and **(c)** decompression for $(4\text{DMAP})_2\text{ZnBr}_4$.

Figure 2. a and b Hirshfeld surface analysis of $(4\text{DMAP})_2\text{ZnBr}_4$ structures at 1at and R-30 GPa. **c** Comparison of absorption spectrum for organic components 4DMAP before compression and after decompression. **d** Illustration of phase transition barrier. ASS, atmospheric stable phase; HPMS, high-pressure metastable state; PB, potential barrier.

References

- 1 Luo, J. et al. Aggregation-induced emission of 1-methyl-1, 2, 3, 4, 5-pentaphenylsilole. *Chem. Commun.*, 1740-1741 (2001).
- 2 Xu, W. et al. Self-Stabilized Amorphous Organic Materials with Room-Temperature Phosphorescence. *Angew. Chem. Int. Ed.* 58, 16018-16022 (2019).

The corresponding revised details are highlighted in yellow and can be found in Line 16-18, 6-7, 16, Page 7, 8, 10 of the Manuscript in the revised manuscript.

Comments 8: *The mechanism of pressure-induced suppression of the molecular rotation could be compared with relevant literature on hybrid (organic-inorganic) compounds (e.g. doi: 10.1021/acs.jpcc.7b11461J, doi: 10.1021/acs.jpcc.2c08253). In HOIPs (hybrid organic inorganic perovskites), for example, the application of pressure drives a transition from a dynamically disordered phase at ambient conditions, where the molecules freely rotate, to a statically disordered phase at high pressure. In the latter case, the molecules are locked at random orientations giving rise to an amorphous-like phase. Remarkably, in HOIPs, this amorphous-like phase is often reversible, meaning that the original crystalline configuration is restored once the sample is brought back at ambient conditions. Does a similar mechanism occur in $(4\text{DMAP})_2\text{ZnBr}_4$? Do the authors have an idea of why, in this case, the transformation is irreversible?*

Author reply: We highly appreciate the reviewer for the insightful comments. We have thoroughly reviewed the literature and found that it delves into the kinetic process in the application of pressure driving a transition to a dynamically disordered phase at ambient conditions, where the molecules rotate freely, into a statically disordered phase at high pressure (doi: 10.1021/acs.jpcc.7b11461J). Meanwhile, similar mechanism occurs in FAPbI_3 (doi: 10.1021/acs.jpcc.2c08253). ABO_3 -type perovskite structures strictly adhere to the Goldschmidt tolerance factor rules, with the large space at the A site enabling free rotation of cations. The small elastic cations, MA^+ and FA^+ , located at the A site in MAPbBr_3 and MAPbI_3 (doi: 10.1021/acs.jpcc.7b11461J and doi: 10.1021/acs.jpcc.2c08253), are locked at random orientations under high pressure, resulting in an amorphous-like phase. However, upon decompression, these

cations rotate freely and return to their initial state at ambient conditions, facilitating the reversibility of the amorphous phase transition once the pressure is fully released.

Low-dimensional hybrid halides do not strictly adhere to the Goldschmidt tolerance factor rules, which grants them significant compositional flexibility.¹ This flexibility allows for the incorporation of larger molecules as A-site cations. In present low-dimensional hybrid halides (4DMAP)₂ZnBr₄, large condensed ring amine cations at A-site, specifically cations 4DMAP⁺, are introduced. The free rotation of these cations is restricted due to the limited space at the A-site and the formed hydrogen bonding interactions both among the large condensed ring amine cations and between these cations and the inorganic Br⁻ components. To investigate the behavior of these molecules under varying pressures, we conducted molecular dynamic simulations. Throughout the simulation process, an ambient temperature of 300 K was maintained as the standard condition (added Supplementary Fig. 16).

Under high pressure, the application of pressure drives a transition from a dynamically disordered phase at ambient conditions to a statically disordered phase at high pressure. As expected by the reviewer, a similar mechanism occurs in (4DMAP)₂ZnBr₄ compared to ABO₃-type perovskite structures. However, it was observed that the cations 4DMAP⁺ rotation angle is limited. Simultaneously, the distances between cations as well as between these cations and Br⁻ gradually decrease. The rigid, large condensed ring amine cations 4DMAP⁺, featured plastic deformation and distorted structure within the (4DMAP)₂ZnBr₄ system (revised Supplementary Fig. 17b and 17d-17f). After high-pressure engineering treatment, the steric hindrance effect resulting from the irreversible plastic deformation, along with distorted large condensed ring amine cations 4DMAP⁺ and the enhanced hydrogen bonding interaction both among the large condensed ring amine cations and between these cations and the inorganic Br⁻ components, are responsible for maintaining the amorphous phase under ambient pressure (revised Supplementary Fig. 17c and 17g-17i).

Added Supplementary Fig. 16. Relationship between temperature and time under different pressures during the molecular dynamic simulation process.

Revised Supplementary Fig. 17. Structural characteristics. Comparison of structures under ambient conditions, high pressure and decompression from different pressures to ambient conditions.

The corresponding revised details are highlighted in yellow and can be found in Line 33-34, 9-17, Page 10, 11 of the Manuscript and Supplementary Figs. 16, 17 of the Supplementary Materials in the revised manuscript.

References

1 Li, X., Hoffman, J. M. & Kanatzidis, M. G. The 2D halide perovskite rulebook: how the spacer influences everything from the structure to optoelectronic device efficiency. *Chem. Rev.* 121, 2230-2291 (2021).

Comments 9: Could the authors provide a concise explanation of what are Hirshfeld calculations? The figures (2h, 2f) reported in the text are obscure for a reader who is not a specialist in the field. What do they describe? What can we learn from them?

Author reply: We sincerely appreciate the reviewer's comment about the Hirshfeld calculations. Hirshfeld calculations have been employed to quantitatively analyze various intermolecular interactions, as well as interactions between these molecular components and inorganic elements within crystal structures, with a particular focus on hydrogen bonding. The fingerprint plot, presented as a color map, offers quantitative insights into the interactions between components within the crystal structure. The Hirshfeld surface, normalized to a reference distance (d_{norm}), serves as a representation of hydrogen bonding interactions. In this representation, red regions indicate the formation of hydrogen bonds, while other colored regions denote weaker interactions compared to hydrogen bonds. The intensity of the red color correlates with the strength of the hydrogen bonding interaction. As illustrated in Fig. 4h, under high pressure, the red color intensifies signify the enhancement of hydrogen bonding interactions. Therefore, under high pressure, the hydrogen bonding interactions are significantly enhanced, both among the large condensed ring amine cations 4DMAP⁺ and between these cations and the inorganic Br⁻ components (Fig. 4f-4h).

Fig. 4. **f** and **g** 2D fingerprint plots of calculated (4DMAP)₂ZnBr₄ structures at 1 atm and 7 GPa. **h** Hirshfeld surface analysis upon compression from 1 atm to 7 GPa.

Comments 10: Could the author explain in detail how the images reported in figure 3 are obtained? What are the substrates used? How is the material deposited on these substrates? How are the patterns created? Is pressure applied before or after the deposition on the substrate? If pressure is applied after the deposition of the material on the substrate, how is the press loaded? The authors should provide much more detail on this procedure and, possibly, also some images of the sample preparation and loading.

Author reply: We sincerely appreciate the reviewer for the insightful comments. We provide a detailed description of the substrate and material deposition process:

Substrate Used: The substrate employed in our experiments was a glass substrate, typically in the form of rectangular glass sheets measuring 20 mm × 50 mm × 0.13 mm. Prior to use, the glass substrate was thoroughly cleaned with acetone, ethanol, and deionized water, followed by a drying process to ensure a clean and contaminant-free surface.

Material Deposition onto Substrate: The target material, $(4\text{DMAP})_2\text{ZnBr}_4$, was first subjected to pressure treatment using a Walker-type large-volume press. This step was carried out in a high-pressure chamber, with the sample typically measuring a few cubic millimeters. After the pressure treatment, the sample was carefully removed and ground into a fine powder. The ground powder was then dispersed in an ethanol solution, with 1 wt% PVP added to ensure uniform dispersion. The dispersed powder was subsequently transferred onto the glass substrate using a direct contact printing technique to create precise micro-patterns.

Pattern Creation: Patterning is accomplished through contact printing. In this process, the dispersed liquid was brought into contact with a mold and substrate, forming a mold-liquid-substrate system. Capillary forces caused the material in the solution to deposit onto the glass substrate according to the mold pattern, resulting in precisely formed micro-patterns. Since the imprinting involves the dispersed liquid, no additional pressure is applied; atmospheric pressure alone is sufficient for pattern formation. The resolution of the pattern depends on the design of the imprinting mold, which is created using laser direct writing or electron beam etching, to form a dispersion-substrate-mold system. This system was left in room temperature for 12 h to allow complete solvent evaporation. During evaporation, capillary forces drove the material to deposit onto the glass substrate, forming precise micron-scale patterns that replicated the mold design. Once the solvent had fully evaporated, the mold was carefully removed, leaving behind the patterned $(4\text{DMAP})_2\text{ZnBr}_4$ on the substrate.

Pressure Loading Process: Firstly, a diamond anvil cell is utilized to investigate the pressure thresholds of the desirable color. However, samples produced by the diamond anvil cell are inadequate for generating anti-counterfeiting patterns. To overcome this limitation, a Walker-type large-volume press is employed for the large mass production of multi-color luminescent samples, which are then utilized for pattern creation. After the high-pressure treatment, the sample was removed from the press chamber and ground into a powder. Pressure application was completed before material deposition onto the substrate, and no additional pressure was applied during the patterning process, which is carried out at atmospheric pressure. Here, the DAC is employed to investigate pressure-induced changes in physical properties, while the Walker-type large-volume press is utilized to prepare large mass samples. The combination of a diamond anvil cell and a Walker-type large-volume press has

proven highly effective in advancing the practical application of metastable optoelectronic functional materials. For instance, this approach has been successfully applied in the fabrication of phosphor-converted light-emitting diodes (pc-LEDs).^{1,2} Notably, using a large-volume press to prepare pressure encryption samples has enhanced their transferability and expanded their practical applications.

In the revised manuscript, we have included details on the substrates used, the material deposition process, pattern creation, and the application of pressure. Micro-Patterning Preparation: The (4DMAP)₂ZnBr₄ sample was loaded to 12 GPa and 21 GPa using a Walker-type large-volume press, followed by decompression to ambient conditions to achieve the desired luminescent properties. After pressure treatment, the modified sample was carefully removed from the press chamber and ground into a fine powder. The powder was then dispersed into an ethanol solution containing 1 wt% PVP to ensure uniform dispersion. The resulting dispersion was drop-coated onto the substrate and brought in contact with a mold, fabricated using laser direct writing or electron beam etching, to form a dispersion-substrate-mold system. This system was left in room temperature for 12 h to allow complete solvent evaporation. During evaporation, capillary forces drove the material to deposit onto the glass substrate, forming precise micron-scale patterns that replicated the mold design. Once the solvent had fully evaporated, the mold was carefully removed, leaving behind the patterned (4DMAP)₂ZnBr₄ on the substrate.

References

1. Xiao, Z. et al. Harvesting Multicolor Photoluminescence in Nonaromatic Interpenetrated Metal–Organic Framework Nanocrystals via Pressure-Modulated Carbonyls Aggregation. *Adv. Mater.* 36, 2403281 (2024).
2. Yang, Q. et al. Pressure treatment enables white-light emission in Zn-IPA MOF via asymmetrical metal-ligand chelate coordination. *Nat. Commun.* 16, 696 (2025).

The corresponding revised details are highlighted in yellow and can be found in Line 27-34, 1-3, Page 16, 17 of the Manuscript in the revised manuscript.

Comments 11: *The introduction section lacks a thoughtful review of the state of the art in the field of PIE luminogen.*

Some relevant aspects, such as the retainability of the photoluminescence signal at ambient conditions and the wavelength range typically spanned, should be discussed. Moreover, in the short paragraph devoted to the discussion of PIE, I found that some of the cited articles seem to be out of topic:

- [16] *Modulating Charge-Density Wave Order and Superconductivity from Two Alternative Stacked*

Monolayers in a Bulk 4Hb-TaSe 2 Heterostructure via Pressure.

- [17] *Pressure-Driven Reversible Switching between n- and p-Type Conduction in Chalcopyrite CuFeS₂*
- [18] *A Protocol to Fabricate Nanostructured New Phase: B31-Type MnS Synthesized under High Pressure*
- [27] *Ultrahard bulk amorphous carbon from collapsed fullerene*
- [28] *Identification of Defect Origin and White-light Emission Tuning of Chalcogenide Quantum Dots through Pressure Engineering. I suggest replacing these articles with more appropriate references or explain better the relevance with the present article.*

Author reply: We have now included a review of the state of the art in the field of PIE luminogen in the introduction section. For instance, in (PEA)₂PbCl₄, pressure engineering enabled the retention of warm white-light emission, with wavelength tuning ranges extending up to 80 nm. Similarly, in (NAPH)₂PbCl₄, the quenched warm white-light emission was achieved, exhibiting a tuning range of 70 nm. However, both materials only showcased two-color emission tuning after applying different pressures. (PEA)₂PbCl₄ transitioned from warm white-light to cold white-light emission, while (NAPH)₂PbCl₄ shifted from cold white-light to warm white-light emission. Furthermore, the objects of PIE luminogens for above-mentioned reports exhibited initially weak emission.

Furthermore, according to the reviewer' suggestions, we have removed the off-topic references (References 16-18, 27, and 28). Additionally, we have included new literatures to enhance the discussions about the influence of hydrostatic and non-hydrostatic pressures, the Urbach tail, and the kinetics of compression on pressure-induced emission retention. These revisions provide a more focused and comprehensive analysis of the relevant mechanisms and phenomena, ensuring a deeper understanding of the system's behavior under varying pressure conditions.

References

- 1 Gaál-Nagy, K. & Strauch, D. Transition pressures and enthalpy barriers for the cubic diamond→ β-tin transition in Si and Ge under nonhydrostatic conditions. *Phys. Rev. B* 73, 134101 (2006).
- 2 Zhuang, Y. et al. Pressure-induced permanent metallization with reversible structural transition in molybdenum disulfide. *Appl. Phys. Lett.* 110 (2017).
- 3 Capitani, F. et al. Locking of methylammonium by pressure-enhanced H-bonding in (CH₃NH₃)PbBr₃ hybrid perovskite. *J. Phys. Chem. C* 121, 28125-28131 (2017).
- 4 Carpenella, V. et al. High-pressure behavior of δ-phase of formamidinium lead iodide by optical spectroscopies. *J. Phys. Chem. C* 127, 2440-2447 (2023).

5 Shang, Y.-C. et al. Pressure Generation above 35 GPa in a Walker-Type Large-Volume Press*. *Chin. Phys. Lett.* 37, 080701 (2020).

6 Li, X., Hoffman, J. M. & Kanatzidis, M. G. The 2D halide perovskite rulebook: how the spacer influences everything from the structure to optoelectronic device efficiency. *Chem. Rev.* 121, 2230-2291 (2021).

7 Pan, Y., Inam, F., Zhang, M. & Drabold, D. A. Atomistic Origin of Urbach Tails in Amorphous Silicon. *Phys. Rev. Lett.* 100, 206403 (2008).

8 Luo, Y. & Flewitt, A. J. Understanding localized states in the band tails of amorphous semiconductors exemplified by a-Si: H from the perspective of excess delocalized charges. *Phys. Rev. B* 109, 104203 (2024).

9 Rambadey, O. V., Kumar, K. & Sagdeo, P. R. Temperature dependence of disorder sensitivity of phonon modes in finite-gap materials. *Phys. Rev. B* 110, 235201 (2024).

10 Shi, K. et al. Sulfur chains glass formed by fast compression. *Nat. Commun.* 16, 357 (2025).

The corresponding revised details are highlighted in yellow and can be found in Line 16-22, Page 3, 19, 20 of the Manuscript in the revised manuscript.

Comments 12: I would recommend a reorganization of the final section of the manuscript, as some repetitions are present in the text.

"Additionally, the fluorescent patterns can carry encrypted information, such as microscopic QR code labels. These labels can be deciphered using microscopic fluorescence imaging, enabling information retrieval from the embedded website information."

[...]

"Additionally, the fluorescent patterns of cyan-blue emission could also carry encrypted information, such as microscopic QR code labels. These labels can be deciphered using microscopic fluorescence imaging, enabling information retrieval from the embedded website information."

Author reply: We have thoroughly revised and reorganized the final section of the revised manuscript. The revised content now reads: "Additionally, the fluorescent patterns can carry encrypted information, such as microscopic QR code labels. These labels can be deciphered using microscopic fluorescence imaging, enabling information retrieval from the embedded website information. In the case of QR code labels displaying green and cyan-blue emission, users can access the homepage of Jilin University and Linköping University with a phone (Fig. 6c)."

The corresponding revised details are highlighted in yellow and can be found in Line 3-7, Page 13 of the Manuscript in the revised manuscript.

Comments 13: I would recommend a reorganization of the figures.

Most of the panels in figure 1 are redundant:

- The information contained in panel c is basically the same as in a and b.
- Panel d is identical to panel 6a.
- Panel e is unnecessary, the same information can be given in the text without a dedicated figure.
- Comments about panels f,g can be found above.
- A panel showing the intensity values for the photoluminescence signal of all the samples R-7.1, ..., R-35.6 would be particularly useful.

A figure is missing which reports the results from ADXRD in the main text.

Author reply: After carefully considering the valuable suggestions provided by the reviewer, we have revised Fig. 1 and included Fig. 2 of the results from ADXRD in the main text accordingly.

Revised Fig. 1. Potential of $(4\text{DMAP})_2\text{ZnBr}_4$ as the PESK application. **a** Normalized PL spectra of $(4\text{DMAP})_2\text{ZnBr}_4$ measured at 1 atm after loading different pressures. **b** Relationship between emission energy and released pressure from different pressure engineering points. **c** Intensity values for the

photoluminescence signal of all the samples R-7.1, ..., R-35.6. **d** and **e** PL spectra of quenched $(4\text{DMAP})_2\text{ZnBr}_4$ as a function of aging time and cycle numbers by releasing from pressure of 21.5 GPa, respectively.

Added Fig. 2. Changes in ADXRD and lattice parameters under high pressure. **a** Powder X-ray diffraction patterns representative of $(4\text{DMAP})_2\text{ZnBr}_4$ under high-pressure. **b** Experimental lattice constants (a, b and c) and volume of $(4\text{DMAP})_2\text{ZnBr}_4$ as a function of pressures. The solid lines represent the Birch-Murnaghan EOS functions fitted to the measured P-V data. **c** Comparison of refinement results for $(4\text{DMAP})_2\text{ZnBr}_4$ at 1 atm, R-21.5 and R-12.2 GPa, respectively. The purple bars indicate the positions of the refined Bragg peaks, while the black line represents the difference between the experimental (red crosswire) and simulated calculated (black solid line) diffraction profiles.

The corresponding revised details are highlighted in yellow and can be found in Figs. 1, 2 of the Manuscript in the revised manuscript.

Response to Reviewer #3

Reviewer #3 (Remarks to the Author):

I co-reviewed this manuscript with one of the reviewer who provided the listed reports. This is part of the Nature Communications initiative to facilitate training in peer review and to provide appropriate recognition for Early Career Researchers who co-review manuscripts.

Response to Reviewer #4

Reviewer #4 (Remarks to the Author):

Comments: *This manuscript proposes a triple-level and physically uncopiable optical anti-counterfeiting modal 0D organic metal hybrid halides by pressure treatment. This is an interesting topic about of halides perovskites. Through detailed experiment measurement and theoretical calculations, the authors report pressure engineering dependence of controllable multicolored emissions upon different pressures releasing, which are attributed to controllable tuning of halogen-to-ligand charge transfer and local excitation implemented by pressure treatment. The manuscript is well written. The results of experiment optical characterization and theory calculations are reasonable. I haven't found errors or inconsistencies between them. I would like to suggest the publication of this manuscript in Nat Commun, however, some comments are given below for the further revisions.*

Comment 1: *In this study, molecular dynamics simulations are employed to illustrate the microstructure under varying pressures. The temperature of the system typically rises when subjected to pressure. Thus, it is crucial to include temperature change data throughout the process of molecular dynamics simulation in order to confirm that the resulting structure corresponds to 300K, thus enhancing consistency with experimental results.*

Author reply: We thank the reviewer for the insightful suggestions. To better elucidate the microstructure evolution, we have further supplemented the temperature change data throughout the process of molecular dynamic simulation in the revised manuscript. During the molecular dynamic simulation process, an ambient temperature of 300 K was maintained as the standard condition. As shown in added Supplementary Fig. 16, under high pressure, the system experiences an initial increase in temperature, followed by a decrease, and ultimately stabilizes at 300 K. Meanwhile, the expanded local area within a time frame of 2 ps is depicted in illustration of Supplementary Fig. 16. It demonstrates that the structure has reached dynamic equilibrium. Therefore, we chose the microstructure that stabilized at 300 K to serve as the model for investigating the optical properties. Moreover, under different pressures, it can be found that the system temperature remains at 300 K after 0.5 ps, which is in good agreement with the experimental conditions. The calculation details are included in the section outlining the calculations in the main text of manuscript. The NPT ensemble simulations were conducted across pressures ranging from 0 GPa to 50 GPa in increments of 10 GPa, each lasting 5 ps. The simulations utilized a time step of 1 fs and were maintained at a temperature of

300 K. Thus, it is reasonable to use this temperature change data in the process of molecular dynamic simulation, which is consistent with experimental results.

Added Supplementary Fig. 16. Relationship between temperature and time under different pressures during the molecular dynamic simulation process.

The corresponding revised details are highlighted in yellow and can be found in Line 29-33, 1-7, Page 17, 18 of the Manuscript and Supplementary Fig. 16 of the Supplementary Materials in the revised manuscript.

Comment 2: Upon decompression, geometric optimization was applied in the calculations to obtain the amorphous microstructures. Based on what basis or reference were the calculation method chosen?

Author reply: According to the previous research published in Nature 2022, 602, 258, the geometry optimization method proves to well simulate the structural changes following pressure release.¹ This approach is deemed effective and reliable in calculations. Thus, in the present study, the employment of the geometry optimization method is adopted. Here, we used the generalized gradient approximation (GGA) of Perdew, Burke, and Ernzerhof (PBE) with a plane-wave basis kinetic energy cutoff of 400 eV. The Brillouin zone was sampled during self-consistent iterations with k-point grids having a spacing of $2\pi \times 0.05/\text{\AA}$, or better. The equilibrium structures were obtained through total energy minimization, with the residual forces on the atoms converged to below 0.005 eV/\AA . Given

that the intermolecular interaction is important for the structural relaxation step, standard DFT-D3 (IVDW=12) was adopted. Geometry optimizations were performed to model the pressure release process, followed by calculations of radial distribution functions (RDF) and oscillator strengths. We have also added the relevant references to first-principles calculations of the revised manuscript.

References

- 1 He, Y. et al. Superionic iron alloys and their seismic velocities in Earth's inner core. *Nature* 602, 258–262 (2022).

The corresponding revised details are highlighted in yellow and can be found in Line 2-6, Page 18 of the Manuscript in the revised manuscript.

Comment 3: Supplementary Figure S12 illustrates the microstructures following different pressure release conditions. It is essential to label these microstructures as subfigures, maintaining consistency with the labeling format utilized in other Figures.

Author reply: We have now labelled these microstructures as subfigures in revised Supplementary Fig. 17, maintaining consistency with the labeling format utilized in other Figures.

Revised Supplementary Fig. 17. Structural characteristics. Comparison of structures under ambient conditions, high pressure and decompression from different pressures to ambient conditions.

The corresponding revised details are highlighted in yellow and can be found in Supplementary Fig. 17 of the Supplementary Materials in the revised manuscript.

Comment 4: The molecular dynamics simulation calculation method should be included in the section outlining the calculations in the main text rather than being relegated to supplementary materials.

Author reply: Following the suggestion from the reviewer, we have included the molecular dynamics simulation calculation method in the section outlining the calculations in the main text. To investigate the evolution of the crystal structure of $(4\text{DMAP})_2\text{ZnBr}_4$ during compression and decompression, ab initio molecular dynamics (AIMD) simulations were conducted using VASP on a $2\times 2\times 1$ supercell comprising 360 atoms along the x and y axes. The Perdew-Burke-Ernzerhof (PBE) version of the generalized gradient approximation (GGA) was employed to describe the exchange-correlation energy functional. A plane-wave representation with a cutoff energy of 400 eV was used for the wavefunction, and a $1\times 1\times 1$ k-point mesh was applied to the supercell. The total energy convergence criterion was set at 10^{-5} eV. The NPT ensemble simulations were carried out at pressures of 0 GPa, 20 GPa, 30 GPa, 40 GPa, and 50 GPa for 5 ps with a time step of 1 fs and a temperature of 300 K. The microstructure when the system temperature is stable at 300K is selected as the model after pressure is applied. Geometry optimizations were performed to model the pressure release process, followed by calculations of radial distribution functions (RDF) and oscillator strengths.

The corresponding revised details are highlighted in yellow and can be found in Line 29-33, 1-7, Page 17, 18 of the Manuscript in the revised manuscript.

Comment 5: Pressure induces changes in the types of excited states, a phenomenon of great interest. However, only partial information regarding these states is presented in Figure 2 and Supplementary Figure S8. It is suggested that the authors determine and report on specific excited states along with their corresponding types to confirm the accuracy of the provided content.

Author reply: In order to confirm the accuracy of the provided content, we have added other excited state types, such as the low-energy excited state 5th, high-energy excited states 20th, and 30th, respectively. For types of excited states at 0 GPa, the high-energy excited state shows halogen-to-ligand charge transfer (HLCT) excitation types (revised Supplementary Fig. 12e, 12f, 12i, 12j, 12m and 12n). The low-energy excited state exhibits the same type as the first excited state, intramolecular charge transfer (ICT) and HLCT types (revised Supplementary Fig. 12a, 12b). However, for 7 GPa, the low-energy 5th excited state shows local excitation (LE) and HLCT excitation types (revised Supplementary Fig. 12a, 12b), similar to the 1th excited state (Fig. 4b), except for the high excited state being HLCT (revised Supplementary Fig. 12g, 12h, 12k, 12l, 12o and 12p). Comparison of excited states calculations at 0 GPa and 7 GPa based on the multiple excited states, indeed, pressure induces changes in the types of excited states.

Revised Supplementary Fig. 12. Calculated wave functions. a-p Hole and electron of 5th,15th, 20th 30th excited states at 1 atm and at 7 GPa, respectively. El.: electron; Ho.: Hole.

The corresponding revised details are highlighted in yellow and can be found in Supplementary Fig.12 of the Supplementary Materials in the revised manuscript.

Comment 6: As illustrated in Figure 2c, what is the reason for the insignificantly low calculated oscillator strength intensity, and which method was employed?

Author reply: In this context, two reasons account for the low calculated oscillator strength. Firstly, under ambient conditions, the material exhibits non-emissive behavior, resulting in extremely low oscillator strength intensity. However, under high pressure, the intensity is enhanced by a factor of 26, resulting in pressure-induced emission. Despite the increase in intensity, the oscillator strength intensity remains relatively subdued at high pressure due to the limited luminescent strength. Secondly, CP2K's time-dependent density functional theory (TD-DFT) implementation is limited to the time-dependent approximation (TDA), also referred to as TDA-DFT. The calculated oscillator strengths exhibit lower accuracy compared to the precise TD-DFT methodology. While there are limitations in terms of accuracy, the overall trend of change does not pose any significant issues. Therefore, the above two reasons are responsible for the low calculated oscillator strength intensity.